# Boosting anti-PD-1 therapy with metformin-loaded macrophage-derived microparticles

Zhaohan Wei[1,8], Xiaoqiong Zhang[1,8], Tuying Yong[1,2,3], Nana Bie[1], Guiting Zhan[1], Xin Li[1], Qingle Liang[1], Jianye Li[1], Jingjing Yu[4], Gang Huang[5], Yuchen Yan[1], Zelong Zhang[1], Bixiang Zhang[4], Lu Gan [1,2,3✉], Bo Huang [6,7✉] & Xiangliang Yang[1,2,3✉]

The main challenges for programmed cell death 1(PD-1)/PD-1 ligand (PD-L1) checkpoint blockade lie in a lack of sufficient T cell infiltration, tumor immunosuppressive micro-environment, and the inadequate tumor accumulation and penetration of anti-PD-1/PD-L1 antibody. Resetting tumor-associated macrophages (TAMs) is a promising strategy to enhance T-cell antitumor immunity and ameliorate tumor immunosuppression. Here, mannose-modified macrophage-derived microparticles (Man-MPs) loading metformin (Met@Man-MPs) are developed to efficiently target to M2-like TAMs to repolarize into M1-like phenotype. Met@Man-MPs-reset TAMs remodel the tumor immune microenvironment by increasing the recruitment of CD8[+] T cells into tumor tissues and decreasing immuno-suppressive infiltration of myeloid-derived suppressor cells and regulatory T cells. More importantly, the collagen-degrading capacity of Man-MPs contributes to the infiltration of CD8[+] T cells into tumor interiors and enhances tumor accumulation and penetration of anti-PD-1 antibody. These unique features of Met@Man-MPs contribute to boost anti-PD-1 antibody therapy, improving anticancer efficacy and long-term memory immunity after combination treatment. Our results support Met@Man-MPs as a potential drug to improve tumor resistance to anti-PD-1 therapy.

[1] National Engineering Research Center for Nanomedicine, College of Life Science and Technology, Huazhong University of Science and Technology, Wuhan 430074, China. [2] Key Laboratory of Molecular Biophysics of the Ministry of Education, College of Life Science and Technology, Huazhong University of Science and Technology, Wuhan 430074, China. [3] Hubei Key Laboratory of Bioinorganic Chemistry and Materia Medica, Huazhong University of Science and Technology, Wuhan 430074, China. [4] Hepatic Surgery Center, Tongji Hospital, Tongji Medical College, Huazhong University of Science and Technology, Wuhan 430030, China. [5] School Hospital, Huazhong University of Science and Technology, Wuhan 430074, China. [6] Department of Immunology & National Key Laboratory of Medical Molecular Biology, Institute of Basic Medical Sciences, Chinese Academy of Medical Sciences, Peking Union Medical College, Beijing 100005, China. [7] Department of Biochemistry and Molecular Biology, Tongji Medical College, Huazhong University of Science and Technology, Wuhan 430030, China. [8] These authors contributed equally: Zhaohan Wei, Xiaoqiong Zhang. ✉email: lugan@mail.hust.edu.cn; tjhuangbo@hotmail.com; yangxl@mail.hust.edu.cn

Programmed cell death 1(PD-1)/PD-1 ligand (PD-L1) checkpoint blockade is a promising clinical anticancer treatment modality by blocking the binding of PD-L1 on tumor cells to PD-1 on activated T cells to reactivate T-cell-mediated antitumor immunity[1–3]. However, the durable response rate to anti-PD-1/PD-L1 therapy remains relatively low in most cases[4,5]. The major factors attributed to the resistance to anti-PD-1/PD-L1 therapy include the lack of infiltrating T lymphocytes in tumor tissues characterizing the so-called "cold tumor"[6], and the presence of tumor immunosuppressive microenvironment, such as regulatory T (Treg) cells and myeloid-derived suppressor cells (MDSCs) to inhibit the anti-PD-1/PD-L1 antibody-regenerated anti-tumor cytotoxic T lymphocytes and Th1 cell response[7,8]. In addition, the aberrant vascular architecture, elevated interstitial fluid pressure, and compact extracellular matrix (ECM) in tumor tissues hamper the tumor accumulation of anti-PD-1/PD-L1 antibody and subsequent deep penetration into tumor parenchyma[9,10], which limits the therapeutic effects of anti-PD-1/PD-L1 antibody. Therefore, efficient T lymphocyte tumor infiltration, improved tumor immunosuppressive microenvironment, and enhanced tumor accumulation and penetration of anti-PD-1/PD-L1 antibody are required to increase the potential responses to anti-PD-1/PD-L1 antibody.

Tumor-associated macrophages (TAMs), one of the most abundant tumor-infiltrating leukocytes in various tumors, are commonly educated by tumor microenvironment to different phenotypes[11,12]. M1-like TAMs, recognized as classically activated macrophages, exhibit anticancer activity by releasing nitrogen oxide (NO) and stimulating naïve T cells to make a Th1/cytotoxic response[13]. In contrast, M2-like TAMs, alternatively activated macrophages predominantly present in tumors, promote tumor growth, angiogenesis, metastasis and tumor immune escape[14]. M2-like TAMs can directly or indirectly inhibit T cell functions by expressing T cell immune checkpoint ligands (such as PD-L1 and PD-L2)[15,16], secreting inhibitory cytokines (such as IL-10 and transforming growth factor-β)[17], and recruiting immunosuppressive cells (such as Treg cells)[18]. Therefore, repolarization of M2-like TAMs to M1 phenotype is a promising strategy to improve the T cell-mediated antitumor immunity and ameliorate the immunosuppressive tumor microenvironment, which might be beneficial for the enhanced anticancer activity of anti-PD-1/PD-L1 antibody.

Metformin (Met), a popular drug used to treat diabetes due to its good hypoglycemic effect, few side effects, and low cost[19], exhibits anticancer activity[20]. A large number of studies have shown that Met can regulate tumor metabolism[21], arrest tumor cell cycle[22], inhibit angiogenesis[23], and kill cancer stem cells by activating adenosine 5′-monophosphate (AMP)-activated protein kinase (AMPK)[24]. Recently, Met has been found to efficiently repolarize M2-like TAMs to M1 phenotype to inhibit tumor growth and metastasis through the AMPK-NF-κB signaling pathway[25,26]. However, how to achieve the targeted delivery of Met to M2-like TAMs to further improve the therapeutic effects remains a big challenge.

Cellular microparticles (MPs) are extracellular vesicles with a diameter of 100–1000 nm that are shed by cells in response to various endogenous or exogenous stimuli[27–29]. MPs hold great potential as drug-delivery systems due to the unique property to deliver messenger molecules, enzymes, and genetic material (DNAs, RNAs) between cells[30–32], and the superior circulation stability, high biocompatibility, low immunogenicity and toxicity[33–35]. Considering that TAMs are derived from monocytes or tissue-resident macrophages recruited by various signals in the tumor microenvironment (such as cytokines, chemokines, ECM components, and hypoxia, etc.)[14,36], macrophage-derived MPs have natural tumor-targeting capacity[37]. M2-like TAMs highly express

mannose receptor CD206/MRC1[38,39]. In this study, mannose (Man)-modified macrophage-derived MPs are used as carriers for targeted delivery of Met (denoted as Met@Man-MPs) to M2-like TAMs. Met@Man-MPs efficiently reset TAMs toward M1 phenotype to inhibit tumor growth. Importantly, Met@Man-MPs significantly improve tumor immunosuppressive microenvironment, and enhance CD8+ T cell infiltration into tumor interiors by the reset macrophages-induced recruitment of CD8+ T cells and Man-MPs-induced tumor ECM degradation since macrophages express matrix metalloproteinases (MMPs)[40,41]. Meanwhile, Met@Man-MPs efficiently enhance tumor accumulation and penetration of anti-PD-1 antibody. These unique features of Met@Man-MPs contribute to the anticancer treatment of anti-PD-1 antibody, resulting in strong anticancer efficacy and the generation of long-term memory immunity after co-administration (Fig. 1a).

## Results

**Preparation and characterization of Met@Man-MPs**. M2-like macrophages highly expressed *Mrc1* gene (Supplementary Fig. 1), which encoded mannose receptor CD206/MRC1. To acquire Man-modified MPs to achieve M2-like macrophage targeting, murine RAW264.7 macrophages were first incubated with DSPE-PEG-Man to obtain Man-engineered cells by virtue of the natural membrane phospholipid exchange of cells[42]. RAW264.7 cells and Man-engineered RAW264.7 cells were then treated with Met (2 mg mL$^{-1}$), followed by ultraviolet irradiation for Met-packaging MPs and Man-MPs (denoted as Met@MPs and Met@Man-MPs, respectively) induction[34,35]. Lectin recognition assay confirmed the successful modification of Man in Man-MPs and Met@Man-MPs (Supplementary Fig. 2). Dynamic light scattering (DLS) analysis showed that Met@MPs and Met@Man-MPs have a similar size of 440 nm (Fig. 1b) and zeta potential of −11.6 mV (Fig. 1c), respectively. Transmission electron microscopy (TEM) revealed that both Met@MPs and Met@Man-MPs were monodisperse and irregularly spherical (Fig. 1d). High-performance liquid chromatography (HPLC) analysis showed that the drug loading capacity for both MPs and Man-MPs was about 0.69 μg of Met per μg protein (MPs and Man-MPs were quantified according to the protein content). Furthermore, Met@MPs and Met@-Man-MPs showed a pH-responsive sustained drug release (Fig. 1e). Met@MPs and Met@Man-MPs did not exhibit significant changes of the size or zeta potentials in phosphate-buffered saline (PBS) with or without 10% fetal bovine serum (FBS) after 7 days (Supplementary Fig. 3), suggesting that Met@MPs and Met@Man-MPs are relatively stable.

**Man-MPs efficiently target to M2-like TAMs**. To investigate the M2-like TAM-targeting capacity of Man-MPs, we first evaluated the uptake efficiency of PKH67-labeled MPs and Man-MPs by RAW264.7 cells (M0 macrophages), LPS- and IFN-γ-conditioned RAW264.7 cells (M1-like macrophages), IL-4-conditioned RAW264.7 cells (M2-like macrophages), murine dendritic DC2.4 cells which are another main phagocytes expressing mannose receptor CD206[43], and murine H22 hepatocarcinoma cells by flow cytometry (Fig. 2a). The strongest intracellular PKH67 fluorescence was detected in M2-like macrophages treated with Man-MPs among all the groups, suggesting a selective M2-like macrophage targeting of Man-MPs. Consistently, more Met@Man-MPs were internalized into M2-like macrophages compared with Met@MPs and free Met (Supplementary Fig. 4). The M2-like macrophage targeting capability of Man-MPs was confirmed in IL-4-conditioned murine bone marrow-derived macrophages (M2-like BMDMs, Fig. 2b). Pretreatment with free Man significantly decreased the intracellular accumulation of

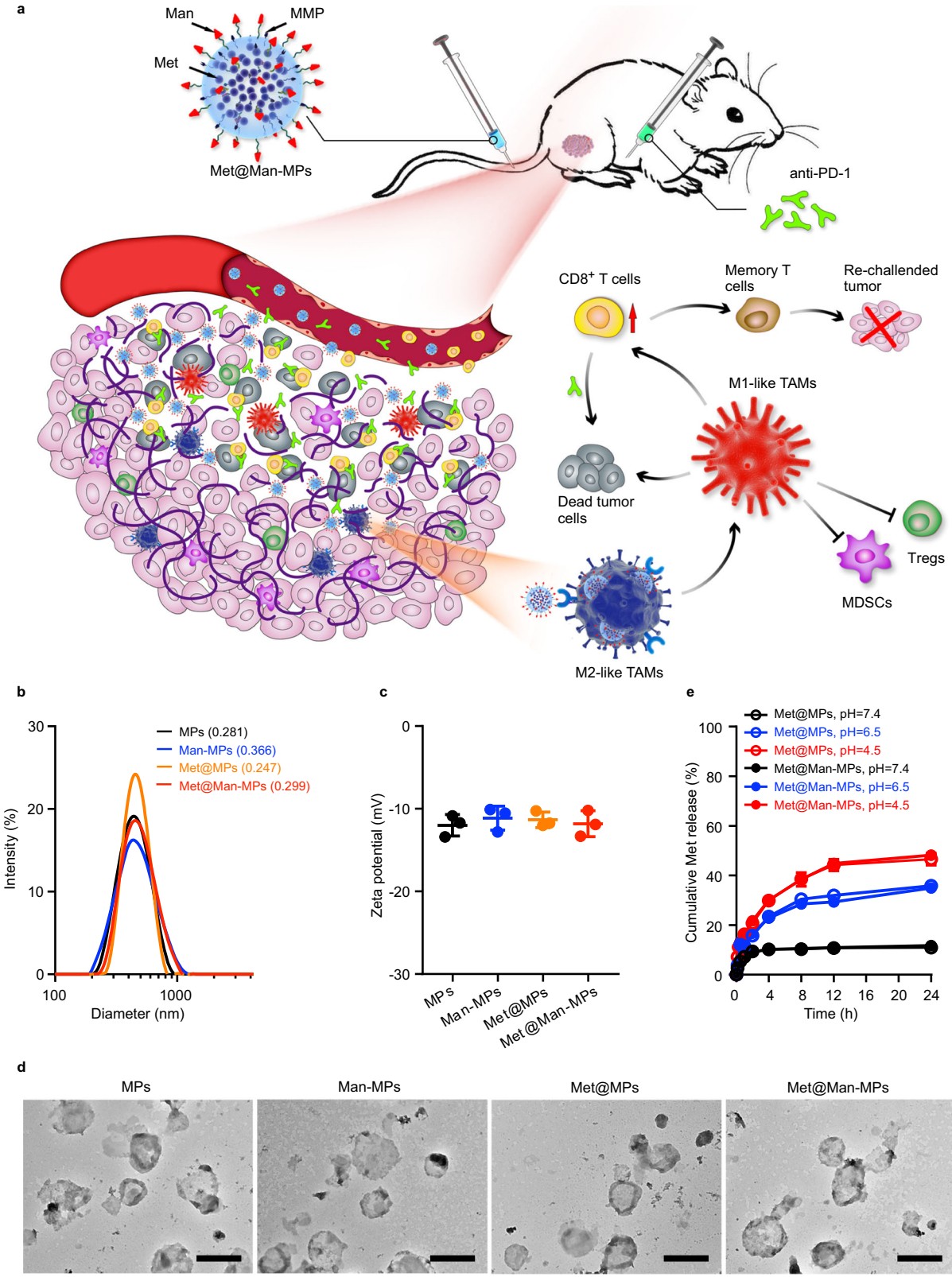

Man-MPs in M2-like macrophages (Fig. 2c), revealing that the M2-like macrophage targeting ability of Man-MPs was mediated by mannose receptor CD206 expressed in M2-like macrophages.

The M2-like TAM-targeting capacity of Man-MPs was further confirmed in vivo. H22 tumor-bearing mice were intravenously injected with Cy5-labeled MPs or Man-MPs. Biodistribution analysis showed that at 24 h after administration, MPs and Man-MPs were mainly accumulated in tumors, revealing the tumor-targeting capacity of macrophages-derived MPs (Fig. 2d, e). However, more Man-MPs were accumulated in tumor tissues than MPs (Fig. 2d, e). Meanwhile, the highest Met accumulation in tumor tissues was also detected in Met@Man-MPs-treated group (Supplementary Fig. 5). Furthermore, green fluorescent protein (GFP)-expressing H22 tumor-bearing mice were

**Fig. 1 Characterization of Met@Man-MPs. a** Schematic of Met@Man-MPs as an efficient drug to boost anti-PD-1 therapy. Met@Man-MPs with MMP activity efficiently target to M2-like TAMs and degrade tumor collagen. (1) Met@Man-MPs repolarize M2-like TAMs to M1-like phenotype, resulting in the recruitment of CD8[+] T cells into tumor tissues and the ameliorated tumor immunosuppressive microenvironment. (2) Collagen-degrading capacity of Man-MPs contributes to the infiltration of CD8[+] T cells into tumor interiors and enhances tumor accumulation and penetration of anti-PD-1 antibody. (3) Met@Man-MPs synergistically inhibit tumor growth in combination with anti-PD-1 antibody, generating long-term memory immunity. **b** Hydrodynamic diameters of MPs and Man-MPs with or without Met by DLS analysis. Polydispersity index values are indicated in the brackets. **c** Zeta potentials of MPs and Man-MPs with or without Met by DLS analysis. Data are presented as means ± s.d. ($n = 3$ independent samples). **d** Morphology of MPs and Man-MPs with or without Met by TEM. Images are representative of three independent experiments. Scale bars: 500 nm. **e** Drug release of Met@MPs and Met@Man-MPs in PBS at different pH values. Data are presented as means ± s.d. ($n = 3$ independent samples). Source data are provided as a Source Data file.

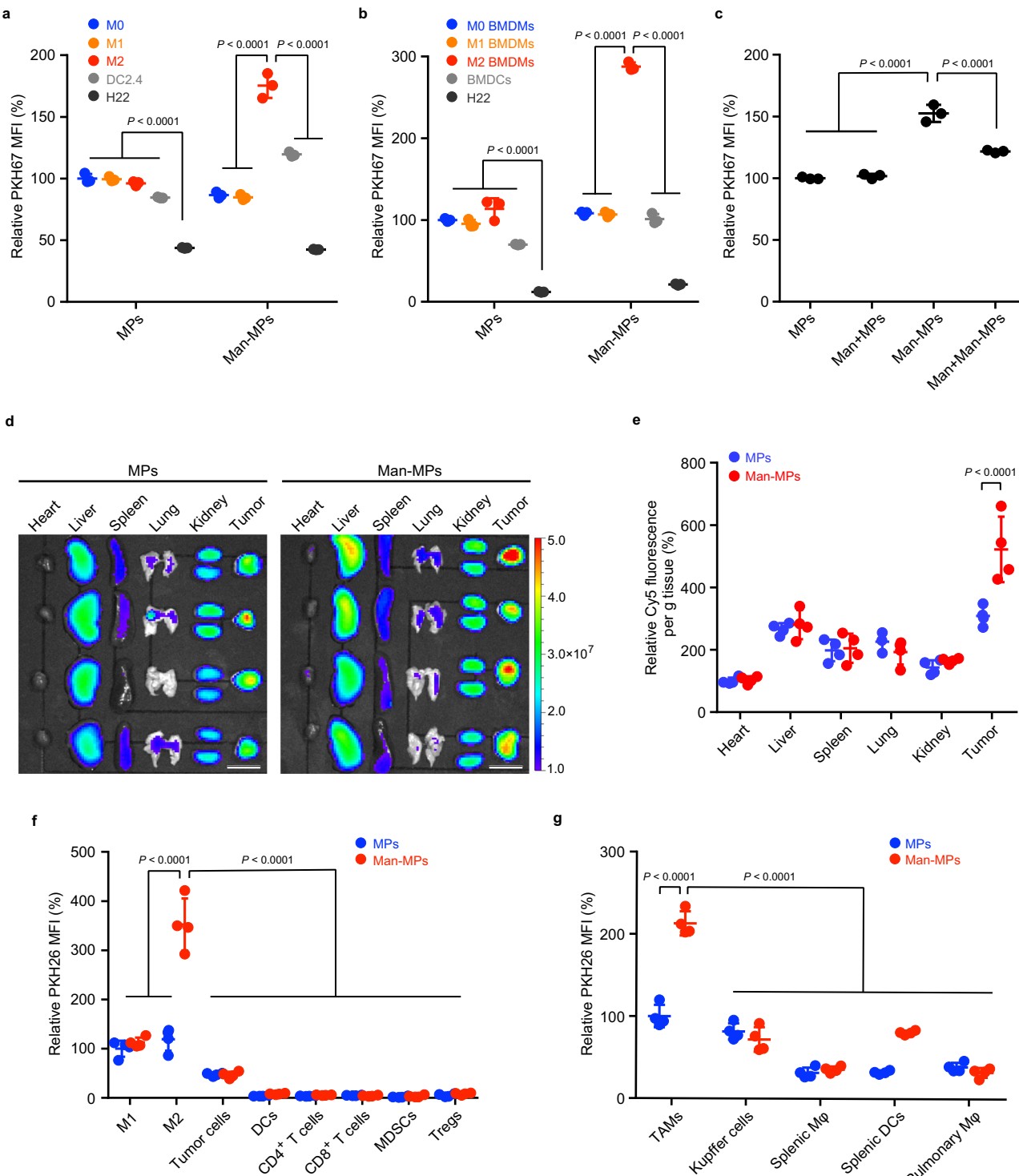

**Fig. 2 M2-like TAM-targeting capacity of Man-MPs. a, b** Relative PKH67 mean fluorescence intensity (MFI) in RAW264.7 cells (M0 macrophages, **a**) or BMDMs (M0 BMDMs, **b**), LPS- and IFN-γ-conditioned RAW264.7 cells (M1-like macrophages, **a**) or BMDMs (M1-like BMDMs, **b**), IL-4-conditioned RAW264.7 cells (M2-like macrophages, **a**) or BMDMs (M2-like BMDMs, **b**), dendritic DC2.4 cells (**a**) or bone marrow-derived dendritic cells (BMDCs, **b**) and H22 cells after treatment with PKH67-labeled MPs or Man-MPs at the concentration of 10 μg protein mL$^{-1}$ for 4 h. Data are presented as means ± s.d. ($n = 3$ biologically independent samples; one-way ANOVA followed by Tukey's HSD post-hoc test). **c** Relative PKH67 MFI in IL-4-conditioned RAW264.7 cells after treatment with PKH67-labeled MPs or Man-MPs at the concentration of 10 μg protein mL$^{-1}$ in the presence or absence of 100 μg mL$^{-1}$ Man for 4 h. Data are presented as means ± s.d. ($n = 3$ biologically independent samples; one-way ANOVA followed by Tukey's HSD post-hoc test). **d, e,** Ex vivo imaging (**d**) and fluorescence intensity (**e**) of Cy5 in the organs and tumors of H22 tumor-bearing mice at 24 h after intravenous injection of Cy5-labeled MPs or Man-MPs at the dosage of 15 mg protein kg$^{-1}$. Scale bars: 1 cm for (**d**). Data are presented as means ± s.d. for (**e**). ($n = 4$ mice per group; unpaired two-tailed Student's $t$-test). **f** Relative PKH26 MFI in M1-like TAMs (CD11b$^+$F4/80$^+$CD11c$^+$ cells), M2-like TAMs (CD11b$^+$F4/80$^+$CD206$^+$ cells), GFP$^+$ tumor cells, DCs (CD45$^+$F4/80$^-$CD11c$^+$ cells), CD4$^+$ T cells (CD45$^+$CD3$^+$CD4$^+$ cells), CD8$^+$ T cells (CD45$^+$CD3$^+$CD8a$^+$ cells), MDSCs (CD45$^+$CD11b$^+$Gr1$^+$ cells) and Tregs (CD45$^+$CD4$^+$CD25$^+$FoxP3$^+$ cells) in tumor tissues of GFP-expressing H22 tumor-bearing mice at 24 h after intravenous injeciton of PKH26-labeled MPs or Man-MPs at the dosage of 15 mg protein kg$^{-1}$. Data are presented as means ± s.d. ($n = 4$ mice per group; two-way ANOVA followed by Bonferroni's multiple comparisons post-test). **g** Relative PKH26 MFI in TAMs, liver Kupffer cells, splenic macrophages, pulmonary macrophages (all these macrophages including TAMs were gated as CD11b$^+$F4/80$^+$ cells) and splenic DCs (CD45$^+$F4/80$^-$CD11c$^+$ cells) of H22 tumor-bearing mice at 24 h after treatment indicated in (**f**). Data are presented as means ± s.d. ($n = 4$ mice per group; two-way ANOVA followed by Bonferroni's multiple comparisons post-test). Source data are provided as a Source Data file.

intravenously injected with PKH26-labeled Man-MPs or MPs. At 24 h after administration, the tumor tissues were digested into single-cell suspensions and the intracellular PKH26 fluorescence in different types of cells was evaluated by flow cytometry (Fig. 2f **and** Supplementary Fig. 6). M2-like TAMs captured a much greater amount of Man-MPs than MPs. In addition, significantly more Man-MPs were enriched in M2-like TAMs compared with those in other cells, including GFP-positive tumor cells, M1-like TAMs, dendritic cells (DCs), CD4$^+$ T cells, CD8$^+$ T cells, MDSCs, and Tregs, suggesting the superior M2-like TAM-targeting capability of Man-MPs. Confocal microscopic images also confirmed that more Man-MPs were colocalized with M2-like TAMs compared with other immune cells in tumor tissues (Supplementary Fig. 7). Due to the tumor-targeting capacity of macrophages-derived MPs and high CD206 expression in TAMs (Supplementary Fig. 8), PKH26 fluorescence in TAMs was significantly higher than that in Kupffer cells, splenic macrophages, splenic DCs, and pulmonary macrophages at 24 h after intravenous injection of PKH26-labeled Man-MPs (Fig. 2g), excluding the targeting accumulation of Man-MPs in other tissue-resident macrophages and phagocytes. Meanwhile, Man-MPs were found to be mainly colocalized with CD206$^+$ cells in tumor tissues (Supplementary Fig. 9) although CD209, a C-type lectin, can also bind mannose residues[44].

**Met@Man-MPs efficiently repolarize M2-like macrophages to M1 phenotype.** The repolarization of M2-like macrophages to M1 phenotype by Met@Man-MPs derived from RAW264.7 cells was evaluated in IL-4-conditioned RAW264.7 cells (Fig. 3a–e and Supplementary Fig. 10), which expressed typical M2-like macrophage markers (Supplementary Fig. 11a). Compared with PBS, MPs, Man-MPs, and free Met, treatment with Met@MPs significantly enhanced the mRNA expression of *TNFα* and *iNOS*, and protein expression of CD80 and CD86 (M1-related markers), while decreased the mRNA expression of *Arg1*, *Mrc1* and *Mgl1*, and protein expression of CD206 (M2-related markers). However, Met@Man-MPs exhibited the strongest M2 macrophage repolarization activity, even better than the high dosage of free Met (Fig. 3a–e and Supplementary Fig. 10), which was confirmed in IL-4-conditioned BMDMs (Supplementary Fig. 12). In addition, Met@Man-MPs derived from BMDMs also showed the excellent M2 macrophage repolarization activity in IL-4-conditioned BMDMs (Supplementary Fig. 13). A similar phenomenon was further confirmed using Met@Man-MPs derived from human monocytic THP-1-originated macrophages and human peripheral blood

monocyte-derived macrophages (MDMs) in IL-4-conditioned THP-1-originated macrophages (Supplementary Fig. 14) and MDMs (Supplementary Fig. 15), respectively. These results suggest that the M2-like macrophage repolarization activity of Met@Man-MPs is universal. Compound C (CC), an AMPK inhibitor significantly abrogated Met@Man-MPs-induced increase in M1-related markers (*TNF-α* and *iNOS*) and decrease in M2-related markers (*Mrc1* and *Arg1*) (Supplementary Fig. 16), suggesting that similar to Met, Met@Man-MPs repolarized M2-like macrophages into M1 phenotype through an AMPK signaling pathway. Caveolin-, clathrin-dependent endocytosis and phagocytosis were involved in the cellular uptake of Man-MPs in M2-like macrophages (Supplementary Fig. 17). Intracellular tracing analysis of Rhodamine B (a model drug)-loaded Man-MPs (RhB@Man-MPs) showed that drug and Man-MPs entered M2-like macrophages together and then colocalized with lysosomes, followed by drug release over time (Supplementary Fig. 18).

The classic pro-inflammatory M1-like macrophages possess antitumor activity by secreting proinflammatory cytokines and reactive nitrogen and oxygen species[13]. To further confirm the M2-like macrophage repolarization activity of Met@Man-MPs, we determined the anticancer effects of the supernatants of IL-4-conditioned RAW264.7 cells pretreated with MPs, Man-MPs, free Met, Met@MPs or Met@Man-MPs, or high dosage of free Met. As expected, the supernatants of M2-like macrophages significantly increased the cell viability of murine hepatocellular carcinoma H22 cells, human hepatocellular carcinoma HepG2 cells, and murine 4T1 breast cancer cells (Supplementary Fig. 11b). However, the supernatants of Met@Man-MPs-educated M2-like macrophages exhibited strong inhibition in the cell viability of these cancer cells compared with those of free Met- or Met@MPs-treated group, even more effective than high dosage of free Met group (Fig. 3f-h). The strong cytotoxicity of the supernatants of Met@Man-MPs-educated M2-like macrophages might be due to the secreted TNF-α, as evidenced by the fact that TNF-α content in the supernatants of Met@Man-MPs-treated M2-like macrophages was significantly higher than other groups (Supplementary Fig. 19a), and treatment with etanercept (Etan), an inhibitor of TNF-α significantly decreased the cytotoxicity of the supernatants of Met@Man-MPs-treated M2 macrophages against H22 (Supplementary Fig. 19b) and 4T1 cells (Supplementary Fig. 19c). Meanwhile, Met@Man-MPs-treated M2-like macrophages exhibited stronger phagocytosis of H22 (Supplementary Fig. 20a) and 4T1 cells (Supplementary Fig. 20b). Furthermore, we subcutaneously co-implanted these educated

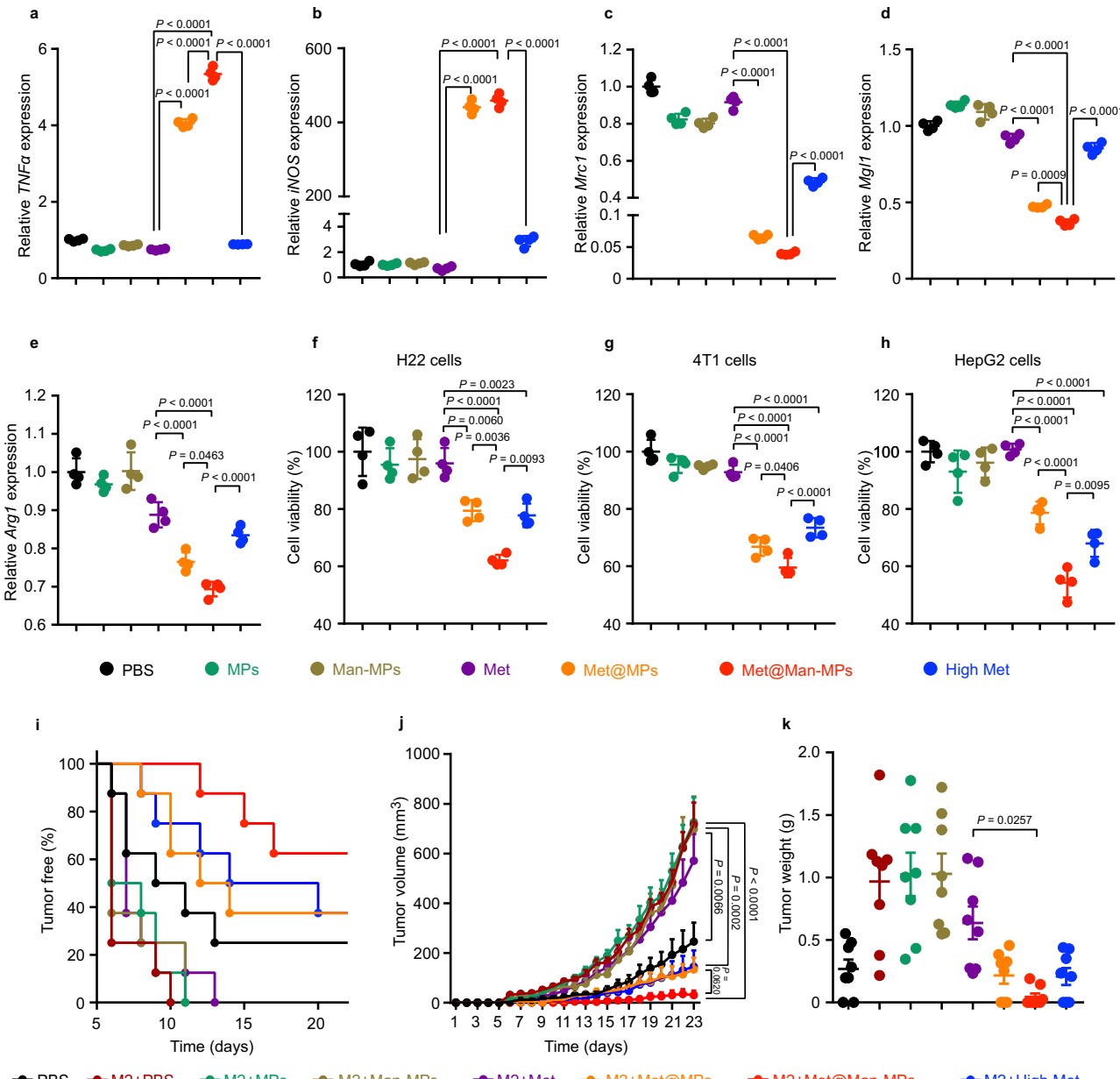

**Fig. 3 Repolarization of M2-like macrophages to M1 phenotype by Met@Man-MPs. a–e** mRNA expression levels of *TNFα* (**a**), *iNOS* (**b**), *Mrc1* (**c**), *Mgl1* (**d**), and *Arg1* (**e**) in IL-4-conditioned RAW264.7 cells after treatment with PBS, MPs, Man-MPs, free Met, Met@MPs or Met@Man-MPs derived from RAW264.7 cells at the Met concentration of 20 μg mL$^{-1}$, or high concentration of Met at 200 μg mL$^{-1}$ for 24 h by real-time RT-PCR. Data are presented as means ± s.d. (*n* = 4 biologically independent samples; one-way ANOVA followed by Tukey's HSD post-hoc test). **f–h** Cell viability of H22 (**f**), 4T1 (**g**), and HepG2 cells (**h**) after treatment with the supernatants of IL-4-conditioned RAW264.7 cells pretreated with PBS, MPs, Man-MPs, free Met, Met@MPs or Met@Man-MPs derived from RAW264.7 cells at the Met concentration of 20 μg mL$^{-1}$, or high dosage of free Met at 200 μg mL$^{-1}$ for 24 h. Data are presented as means ± s.d. (*n* = 4 biologically independent samples; one-way ANOVA followed by Tukey's HSD post-hoc test). **i** Tumor formation in BALB/ c mice after subcutaneous co-implantation of H22 cells (8 × 10$^5$ cells per mouse) and IL-4-conditioned RAW264.7 cells pretreated with PBS, MPs, Man-MPs, free Met, Met@MPs or Met@Man-MPs derived from RAW264.7 cells at the Met concentration of 20 μg mL$^{-1}$, or high dosage of free Met at 200 μg mL$^{-1}$ for 24 h at a ratio of 1:2 into the flanks of BALB/c mice (*n* = 8 mice per group). **j** Tumor volume in BALB/c mice after treatment indicated in (**i**). Data are presented as means ± s.e.m. (*n* = 8 mice per group; two-way ANOVA followed by Bonferroni's multiple comparisons post-test). **k** Tumor weight at the end of treatment indicated in (**i**). Data are presented as means ± s.d. (*n* = 8 mice per group; one-way ANOVA followed by Tukey's HSD post-hoc test). Source data are provided as a Source Data file.

M2-like macrophages and H22 cells into the flanks of BALB/c mice and then assessed the potential of tumor formation. Consistently, the lowest tumor formation ratio (Fig. 3i), the slowest tumor growth (Fig. 3j) and the smallest tumor weight (Fig. 3k) were observed in the group of co-transplantation of Met@Man-MPs-treated M2-like macrophages and H22 cells.

Collectively, these data indicate that Met@Man-MPs efficiently repolarize M2-like macrophages to M1 phenotype.

**Met@Man-MPs significantly inhibit tumor growth and ameliorate tumor immune microenvironment.** To determine the

biological function of Met@Man-MPs, the in vivo antitumor activity of Met@Man-MPs was determined in BALB/c mice bearing subcutaneous H22 tumors. Free Met, Met@MPs or Met@Man-MPs at Met dosage of 10 mg kg$^{-1}$, or high dosage of Met at 100 mg kg$^{-1}$ were intravenously injected into H22 tumor-bearing mice once every other day for six times. Compared with free Met (10 mg kg$^{-1}$), Met@MPs significantly reduced the tumor growth, with 41.6 and 39.9% reduction in tumor volume and weight (Fig. 4a, b). However, Met@Man-MPs exhibited the strongest anticancer activity, achieving 70.2 and 63.2% reduction in tumor volume and weight, even better than a high dosage of Met (Fig. 4a, b). Kaplan–Meier survival analysis showed that 86% of mice were still alive in the Met@Man-MPs-treated group after 63 d, much higher than Met@MPs- and high dosage of Met-treated groups (Fig. 4c). Hematoxylin-eosin (H&E) staining (Supplementary Fig. 21a) and TUNEL analysis (Supplementary Fig. 21b, c) of tumor tissues revealed the minimum number of tumor cells and the maximum number of apoptotic cells in the Met@Man-MPs-treated group, respectively, further confirming the good anticancer activity of Met@Man-MPs. Similar anticancer activity of Met@Man-MPs was found in mice bearing orthotopic 4T1 breast tumors. Met@Man-MPs exhibited the strongest capability to reduce tumor growth (Supplementary Fig. 22a) and prolong the survival time (Supplementary Fig. 22b) compared with free Met, Met@MPs, and high dosage of Met. Moreover, significantly fewer lung metastatic nodules were detected in the Met@Man-MPs-treated group by counting nodule numbers (Supplementary Fig. 22c) and H&E staining analysis of lungs (Supplementary Fig. 22d). In addition, body weight change (Supplementary Fig. 23), H&E staining of major organs (Supplementary Fig. 24), and serological analysis (Supplementary Fig. 25) showed that Met@Man-MPs did not cause obvious toxicity, although high dosage of Met induced nephrotoxicity (Supplementary Figs. 24 and 25).

The tumor immune microenvironment was then investigated in H22 tumor-bearing mice after intravenous injection of free Met, Met@MPs or Met@Man-MPs at Met dosage of 10 mg kg$^{-1}$, or high dosage of Met at 100 mg kg$^{-1}$ once every other day for six times. Consistently, Met@Man-MPs exhibited the strongest ability to increase the number of M1-like TAMs (Fig. 4d) and decrease the number of M2-like TAMs (Fig. 4e) in tumor tissues. Moreover, Met@Man-MPs significantly enhanced the numbers of CD8$^+$ T cells (Fig. 4f), activated CD8$^+$CD69$^+$ T cells (Fig. 4g), CD4$^+$ T cells (Fig. 4h), and activated CD4$^+$CD69$^+$ T cells (Fig. 4i), while markedly decreased the numbers of MDSCs (Fig. 4j) and Tregs (Fig. 4k) in tumor tissues compared with free Met, Met@MPs and even high dosage of Met. Meanwhile, Met@Man-MPs treatment markedly increased the levels of pro-inflammation cytokines IFN-γ (Fig. 4l), TNF-α (Fig. 4m), and IL12p70 (Fig. 4n), while decreased the levels of anti-inflammation cytokine TGF-β (Fig. 4o) in tumor tissues. The improved tumor immune microenvironment induced by Met@Man-MPs was also confirmed in 4T1 tumor-bearing mice (Supplementary Fig. 26a–h), suggesting that Met@Man-MPs efficiently enhance the antitumor immunity and ameliorate the tumor immunosuppression.

To determine whether macrophages were involved in the Met@Man-MPs-mediated antitumor effect, we used clodronate liposomes to deplete macrophages in H22 tumor-bearing mice. As expected, macrophages were effectively depleted by clodronate liposomes (Supplementary Fig. 27a), whereas other immune cells, such as CD8$^+$ T cells (Supplementary Fig. 27b), CD8$^+$CD69$^+$ T cells (Supplementary Fig. 27c), CD4$^+$ T cells (Supplementary Fig. 27d), CD4$^+$CD69$^+$ T cells (Supplementary Fig. 27e), MDSCs (Supplementary Fig. 27f) and Tregs (Supplementary Fig. 27g) were not affected. The macrophage depletion disrupted the

antitumor effect of Met@Man-MPs (Supplementary Fig. 27h, i). Correspondingly, the improved tumor immune microenvironment induced by Met@Man-MPs was abrogated after macrophage depletion (Supplementary Fig. 27b–g), suggesting that the antitumor effect and improved tumor immune microenvironment of Met@Man-MPs is macrophage-dependent. It is well known that M1-like macrophages have the ability to recruit T cells and promote Th1 responses[45]. The improved efficacy following macrophage depletion is often dependent upon enhanced recruitment or function of cytotoxic CD8$^+$ T cells[46,47]. Depletion of CD8$^+$ T cells using anti-CD8 antibody markedly impaired the anticancer ability of Met@Man-MPs (Supplementary Fig. 28a, b). Furthermore, Met@Man-MPs did not exhibit significant tumor inhibition in H22 tumor-bearing nude mice (Supplementary Fig. 28c, d), suggesting that CD8$^+$ T cells are involved in the anticancer activity of Met@Man-MPs.

**Man-MPs efficiently degrade tumor collagen**. Tumor ECM, consisting of a collagen network and proteoglycans, provides a natural barrier against monocyte infiltration and therapeutic drug accumulation and penetration, resulting in the immune evasion and resistance of antitumor therapy[48]. Similar to RAW264.7 macrophages, MPs and Man-MPs derived from RAW264.7 macrophages overexpressed MMP9 and MMP14 proteins (Supplementary Fig. 29a). Meanwhile, MPs and Man-MPs exhibited high MMP activity as detected by the MMP activity assay kit, and loading Met did not affect their MMP activity (Supplementary Fig. 29b). In addition, pretreatment with batimastat (Bati, a broad-spectrum MMP inhibitor) significantly decreased the MMP activity of Met@Man-MPs (Supplementary Fig. 29b), further confirming the MMP activity of Man-MPs. The high MMP activity was also detected in Man-MPs derived from THP-1-originated macrophages (Supplementary Fig. 29c). To evaluate whether MPs and Man-MPs can degrade tumor ECM by MMP activity, PKH26-labeled MPs or Man-MPs pretreated with or without Bati or 4-(2-Aminoethyl) benzenesulfonyl fluoride hydrochloride (AEBSF, an inhibitor of serine proteases, such as plasminogen activation system involved in the activation of a number of MMPs[49]) were incubated with collagen I film. Collagen I was efficiently degraded by MPs or Man-MPs as evidenced by no colocalization between collagen I and MPs or Man-MPs (Supplementary Fig. 30). Pretreatment with Bati or AEBSF significantly decreased MPs- and Man-MP-induced collagen I degradation (Supplementary Fig. 30), suggesting that MMP was involved in the MPs- and Man-MPs-induced collagen degradation. The collagen content determination also supported that MPs and Man-MPs efficiently degraded collagen in a concentration- (Supplementary Fig. 31) and MMP-dependent manner (Fig. 5a). Furthermore, H22 tumor-bearing mice were intravenously injected with PBS, MPs, Man-MPs, free Met, Met@MPs or Met@Man-MPs at the Met dosage of 10 mg kg$^{-1}$, or high dosage of Met at 100 mg kg$^{-1}$ every other day for different times. Second-harmonic generation (SHG) microscopy was used to analyze the fibrillar collagen structure in tumors (Fig. 5b, c). A gradual increase in the SHG signal of tumors was observed in the PBS- and free Met-treated groups. However, MPs and Man-MPs significantly decreased the deposition of collagen fiber network, revealing the collagen-degrading capability of MPs and Man-MPs. Consistently, loading Met into MPs or Man-MPs did not affect their collagen-degrading activity. Immunofluorescence analysis of collagen I in H22 tumor tissues further confirmed that MPs, Man-MPs, Met@MPs, and Met@Man-MPs significantly decreased the collagen area in tumor tissues (Fig. 5d, e).

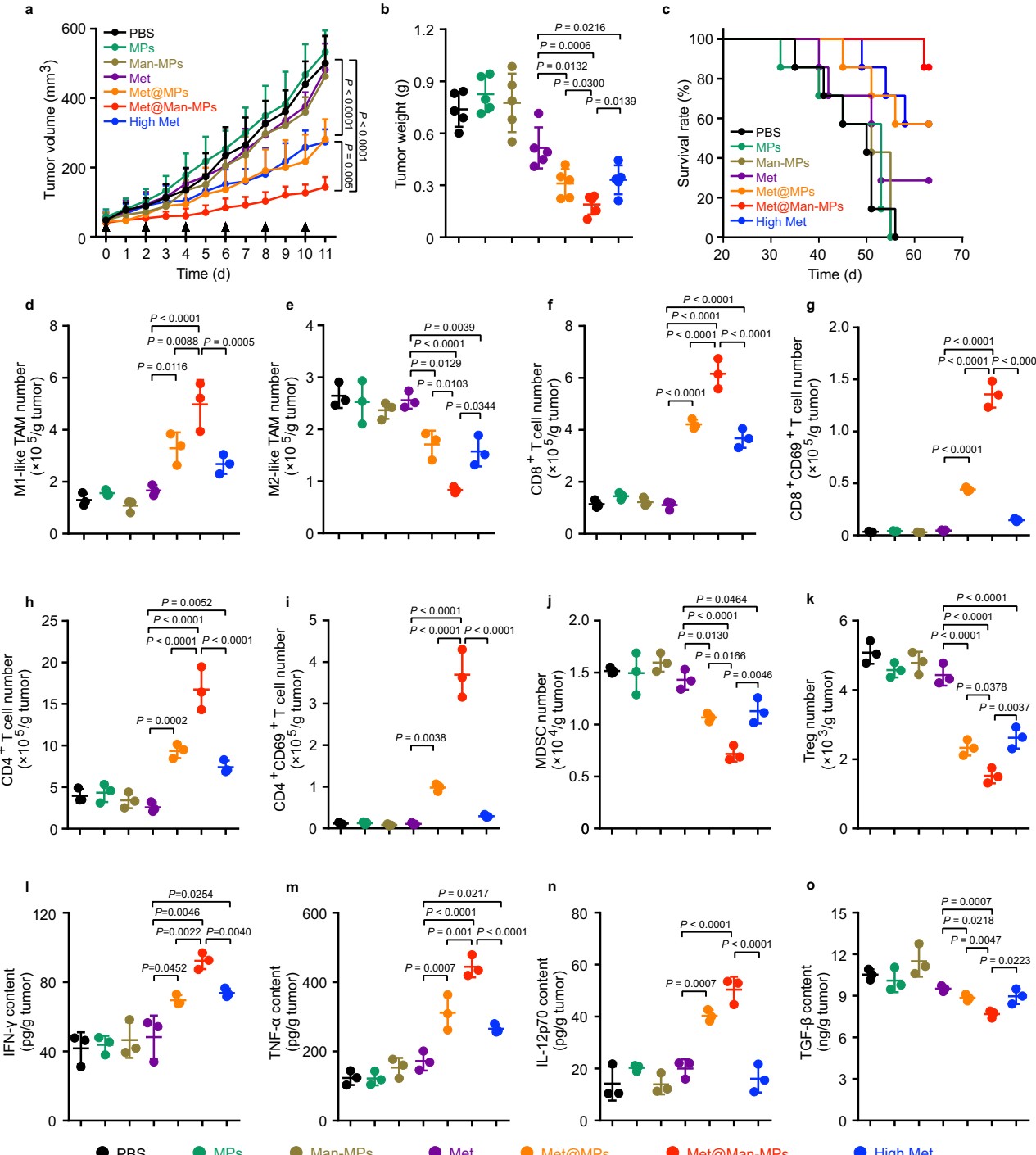

**Fig. 4 Anticancer activity and improved immune microenvironment of Met@Man-MPs in H22 tumor-bearing mice. a** Tumor growth curves of H22 tumor-bearing mice after intravenous injection of PBS, MPs, Man-MPs, free Met, Met@MPs or Met@Man-MPs at the Met dosage of 10 mg kg$^{-1}$, or high dosage of Met at 100 mg kg$^{-1}$ every two days for six times. The black arrows indicate the injection time. Data are presented as mean ± s.e.m. ($n = 5$ mice per group; two-way ANOVA followed by Bonferroni's multiple comparisons post-test). **b** Tumor weight at the end of the treatment indicated in (**a**). Data are presented as mean ± s.d. ($n = 5$ mice per group; one-way ANOVA followed by Tukey's HSD post-hoc test). **c** Kaplan–Meier survival plot of H22 tumor-bearing mice after treatment indicated in (**a**) ($n = 7$ mice per group). **d**–**k** The numbers of M1-like TAMs (**d**), M2-like TAMs (**e**), CD8$^+$ T cells (**f**), CD8$^+$CD69$^+$ T cells (**g**), CD4$^+$ T cells (**h**), CD4$^+$CD69$^+$ T cells (**i**), MDSCs (**j**), and Tregs (**k**) in tumor tissues of H22 tumor-bearing mice after intravenous injection of PBS, MPs, Man-MPs, free Met, Met@MPs or Met@Man-MPs at the Met dosage of 10 mg kg$^{-1}$, or high dosage of Met at 100 mg kg$^{-1}$ every two days for six times. Data are presented as mean ± s.d. ($n = 3$ mice per group; one-way ANOVA followed by Tukey's HSD post-hoc test). **l**–**o** The contents of IFN-γ (**l**), TNF-α (**m**), IL12p70 (**n**), and TGF-β (**o**) in tumor tissues of H22 tumor-bearing mice after intravenous injection of PBS, MPs, Man-MPs, free Met, Met@MPs or Met@Man-MPs at Met dosage of 10 mg kg$^{-1}$, or high dosage of Met at 100 mg kg$^{-1}$ every two days for six times. Data are presented as mean ± s.d. ($n = 3$ mice per group; one-way ANOVA followed by Tukey's HSD post-hoc test). Source data are provided as a Source Data file.

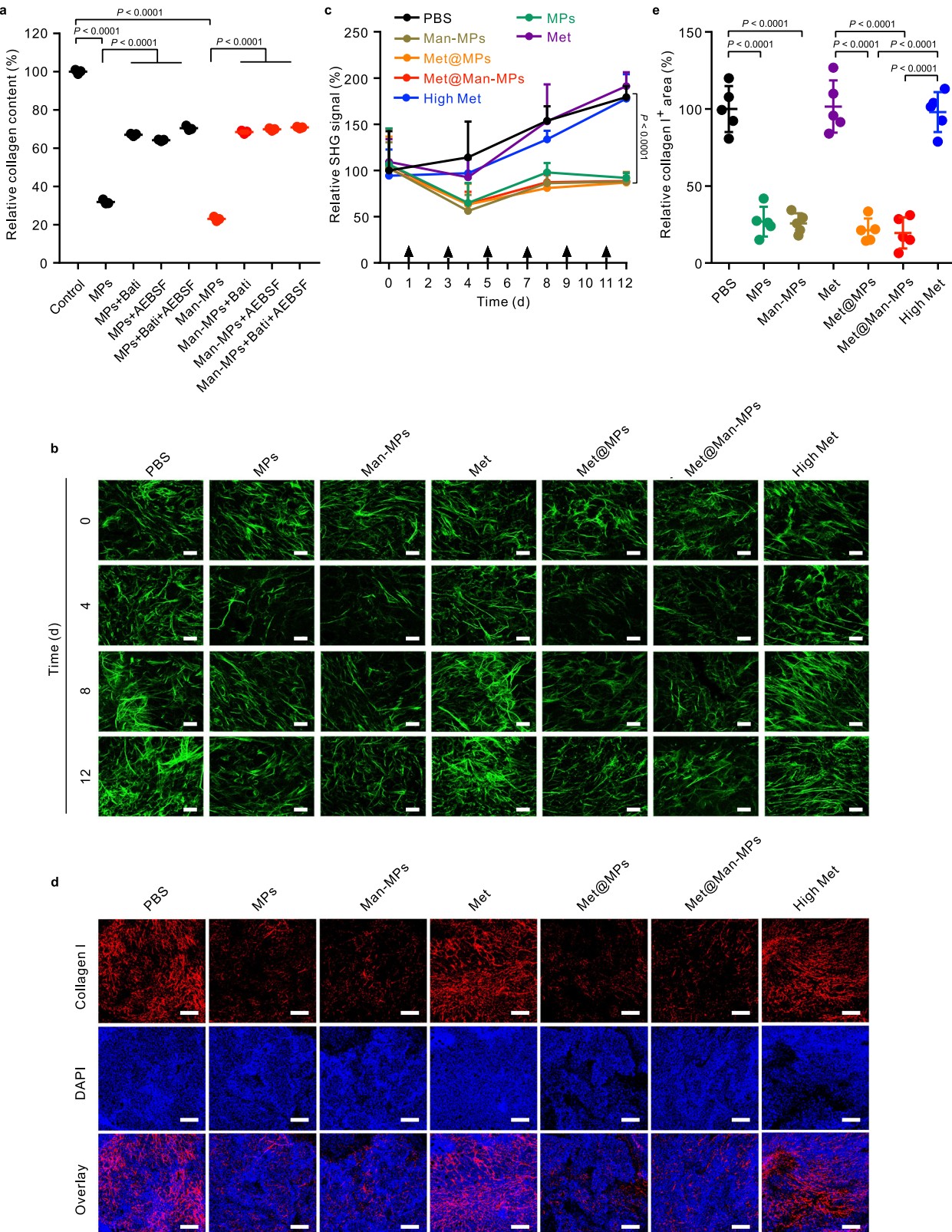

**Met@Man-MPs induce CD8+ T cell infiltration into tumors by recruiting CD8+ T cells and degrading ECM.** CD8+ T cells are recruited by M1-like macrophages by secreting some chemokines and cytokines, such as CXCL-9, CXCL-10, CXCL-11, and TNF-α[50–52]. Transwell analysis showed that Met@Man-MPs-treated M2-like macrophages can efficiently recruit CD8+ T cells (Supplementary Fig. 32). However, TNF-α inhibitor Etan significantly decreased the recruitment of CD8+ T cells by Met@Man-MPs-treated M2-like macrophages (Supplementary Fig. 32), suggesting that TNF-α, upregulated expression and secretion by Met@Man-MPs-treated M2-like macrophages, is responsible for the recruitment of CD8+ T cells. Besides the

**Fig. 5 Collagen-degrading capacity of Man-MPs. a** Relative collagen content after soluble collagen I was incubated with MPs or Man-MPs at the concentration of 75 μg protein mL$^{-1}$ pretreated with or without 0.6 mg mL$^{-1}$ Bati or 100 μg mL$^{-1}$ AEBSF for 72 h measured by Sirius red total collagen detection kit. Data are presented as mean ± s.d. ($n = 3$ biologically independent samples; one-way ANOVA followed by Tukey's HSD post-hoc test). **b** SHG imaging of tumor tissues of H22 tumor-bearing mice after intravenous injection of PBS, MPs, Man-MPs, free Met, Met@MPs or Met@Man-MPs at the Met dosage of 10 mg kg$^{-1}$, or high dosage of Met at 100 mg kg$^{-1}$ every two days for different times. Images are representative of three biologically independent mice. Scale bars: 50 μm. **c** Relative SHG signal intensity in tumor tissues of H22 tumor-bearing mice after treatment indicated in (**b**). Data are presented as mean ± s.d. ($n = 10$ fields for 3 mice; two-way ANOVA followed by Bonferroni's multiple comparisons post-test). **d** Immunofluorescence analysis of collagen I (labeled with Cy3-conjugated collage I antibody, red) in tumor tissues of H22 tumor-bearing mice after intravenous injection of PBS, MPs, Man-MPs, free Met, Met@MPs or Met@Man-MPs at the Met dosage of 10 mg kg$^{-1}$, or high dosage of Met at 100 mg kg$^{-1}$ every two days for six times. The nuclei were stained by DAPI (blue). Images are representative of three biologically independent mice. Scale bars: 200 μm. **e** Relative collagen I-positive area in tumor tissues of H22 tumor-bearing mice after treatment indicated in (**d**). Data are presented as mean ± s.d. ($n = 5$ fields for 3 mice; one-way ANOVA followed by Tukey's HSD post-hoc test). Source data are provided as a Source Data file.

recruitment of CD8$^+$ T cells into tumor tissues, degrading tumor ECM contributes to the infiltration of CD8$^+$ T cells into tumor parenchyma[53]. Immunofluorescence analysis showed that Met@Man-MPs significantly enhanced the CD8$^+$ T cell number in the interior of tumor tissues (Fig. 6b, c). To determine whether Met@Man-MPs-induced CD8$^+$ T cell infiltration was due to the recruitment by the repolarization of M2-like TAM to M1 phenotype and the degradation of tumor ECM, H22 tumor-bearing mice were intravenously injected with Met@Man-MPs pretreated with or without Bati or/and intraperitoneal injection of Etan (Fig. 6a). Pretreatment with Bati significantly decreased the Met@Man-MPs-induced collagen degradation in tumor tissues by SHG microscopy (Supplementary Fig. 33a, b), and Etan treatment significantly decreased TNF-α content in tumor tissues (Supplementary Fig. 33c). Etan or pretreatment with Bati significantly abrogated the Met@-Man-MPs-induced CD8$^+$ T cell increase in the interiors of tumor tissues (Fig. 6b, c). In addition, Met@Man-MPs markedly increased the numbers of CD8$^+$ T (Fig. 6d), CD8$^+$CD69$^+$ T (Fig. 6e), and CD8$^+$IFN-γ$^+$ T cells (Fig. 6f) in tumor tissues by flow cytometry. Etan or pretreatment with Bati markedly decreased the Met@Man-MPs-induced increase in the numbers of these immune cells (Fig. 6d–f). The lowest numbers of these immune cells were detected in Bati-preincubated Met@Man-MPs- and Etan-treated group (Fig. 6d–f). Correspondingly, the anticancer activity of Met@Man-MPs was significantly decreased after pretreatment with Bati/and Etan (Fig. 6g, h). These results suggest that Met@Man-MPs-induced recruitment of CD8$^+$ T cells by the repolarization of M2-like TAMs to M1 phenotype and degradation of tumor ECM are responsible for the increased CD8$^+$ T cell infiltration in the interior of tumor tissues, resulting in the improved anticancer activity of Met@-Man-MPs.

**Met@Man-MPs efficiently enhance tumor accumulation and penetration of anti-PD-1 antibody**. The condensed ECM in tumor tissues critically affects the tumor accumulation and penetration of anti-PD-1 antibody[54]. On the basis of the collagen-degrading capability of Man-MPs, the effects of Met@Man-MPs on the tumor accumulation of anti-PD-1 antibody was assessed in H22 tumor-bearing mice. The mice were intravenously injected with PBS or Met@Man-MPs pretreated with or without Bati every other day for two times, followed by intraperitoneal injection of Cy5-conjugated anti-PD-1 antibody, and the Cy5 fluorescence in different tissues was determined at 24 h after injection (Fig. 7a, b). Met@Man-MPs significantly increased the tumor accumulation of anti-PD-1 antibody. However, pretreatment with Bati abrogated the Met@Man-MPs-induced tumor accumulation of anti-PD-1 antibody, suggesting that the collagen degradation capability of Met@Man-MPs is responsible for the enhanced tumor accumulation of anti-PD-1 antibody.

Next we evaluated the effects of Met@Man-MPs on the tumor penetration of anti-PD-1 antibody. H22 tumor-bearing mice were intravenously injected with PBS or Met@Man-MPs pretreated with or without Bati every other day for two times, followed by intraperitoneal injection of Cy5-conjugated anti-PD-1 antibody. At 24 h after injection, the tumor tissues were stained with fluorescein isothiocyanate (FITC)-conjugated CD31 antibody and then analyzed by confocal microscopy (Fig. 7a, c, d). Anti-PD-1 antibody was strongly colocalized with CD31-labeled tumor vessels due to the dense tumor ECM, although the tumor vessels were permeable as evidenced by the fact that endogenous IgG was distributed throughout the entire tumor tissues (Supplementary Fig. 34). Met@Man-MPs treatment resulted in the wide distribution of anti-PD-1 antibody in tumor tissues, suggesting that anti-PD-1 antibody was extravasated from tumor vessels and then penetrated into tumor tissues. However, the improved tumor penetration of anti-PD-1 antibody was compromised in Bati-pretreated Met@Man-MPs group, revealing that the tumor penetration of anti-PD-1 antibody was achieved by the Met@-Man-MPs-induced collagen degradation in tumor tissues. Furthermore, the dorsal skin window chamber tumor model was constructed to directly observe the extravasation and tumor penetration of anti-PD-1 antibody (Fig. 7e, f). When the tumors were formed, the mice were intravenously injected with PBS or Met@Man-MPs pretreated with or without Bati every other day for two times. The tumor vessels were labeled by intravenous injection of fluorescein-conjugated tomato lectin, followed by intravenous injection of Cy5-conjugated anti-PD-1 antibody. The real-time observation showed that although anti-PD-1 antibody was mainly retained within or around blood vessels, treatment with Met@Man-MPs resulted in the quick extravasation of anti-PD-1 antibody from the blood vessels and subsequent distribution in the extravascular region over time. However, pretreatment with Bati significantly decreased the Met@Man-MPs-induced extravasation and tumor penetration of anti-PD-1 antibody, confirming that Met@Man-MPs efficiently enhance the tumor penetration of anti-PD-1 antibody by collagen degradation in tumor tissues.

**Met@Man-MPs efficiently improve the anticancer activity of anti-PD-1 antibody**. In view of the reshaped tumor immune microenvironment, such as the enrichment of CD8$^+$ T cells, the downregulation of the immunosuppressive cells, the upregulated PD-L1 expression (Supplementary Fig. 35), and the enhanced tumor accumulation and penetration of anti-PD-1 antibody by Met@Man-MPs, the effects of Met@Man-MPs on the anticancer activity of anti-PD-1 antibody were first evaluated in H22 tumor-bearing mice (Fig. 8a–i). As expected, treatment with Met@Man-MPs or anti-PD-1 antibody alone exhibited significant anticancer activity. Man-MPs treatment significantly increased the anticancer activity of anti-PD-1 antibody, which might be due to the enhanced

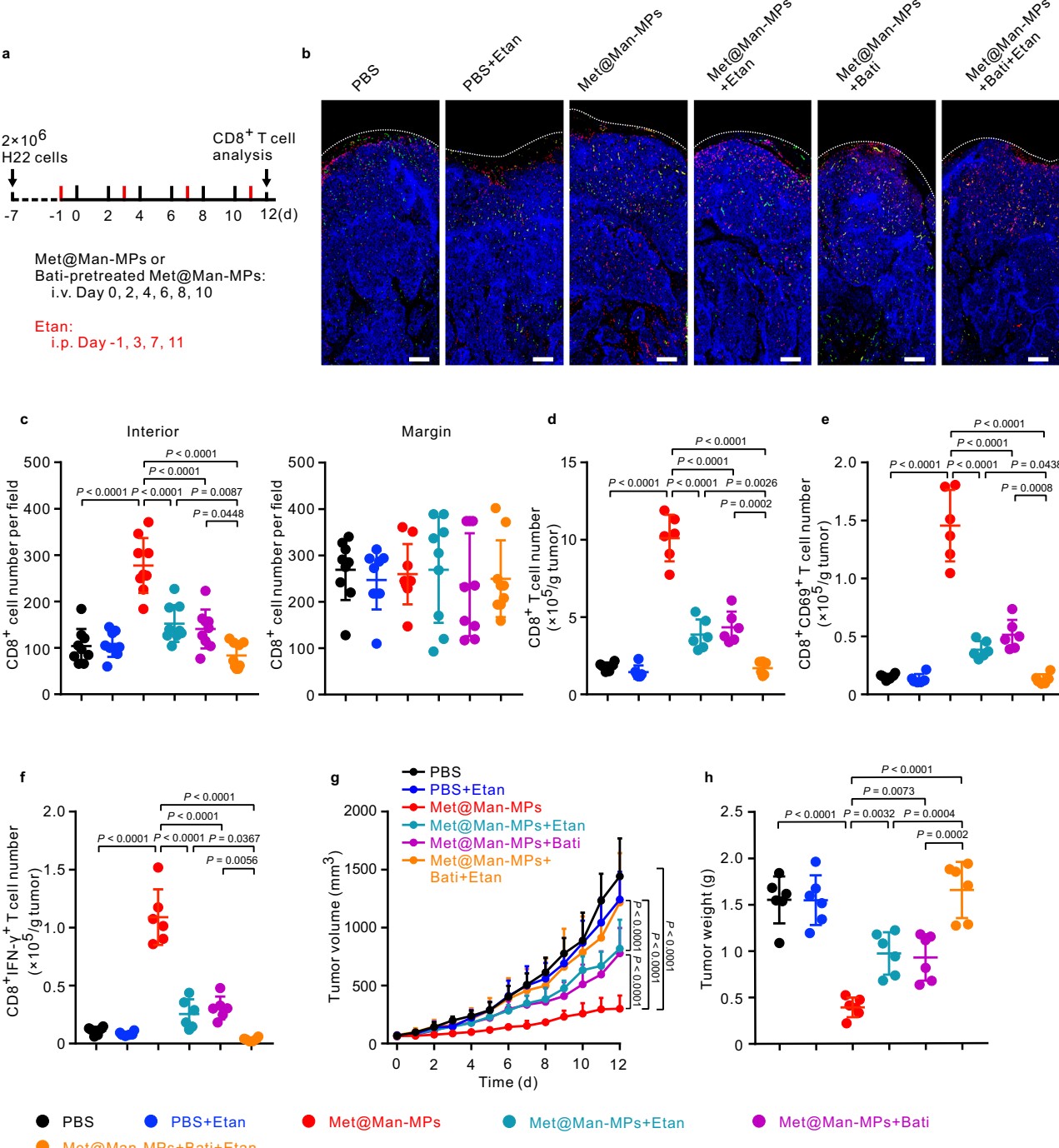

**Fig. 6 Met@Man-MPs-induced CD8+ T cell tumor infiltration by recruiting CD8+ T cells and degrading tumor ECM. a** Schematic schedule for Met@Man-MPs-induced CD8+ T cell tumor infiltration analysis in H22 tumor-bearing mice. **b** Fluorescence images of CD8+ T cells (labeling with Cy3-conjugated CD8 antibody, red) in the tumor tissues of H22 tumor-bearing mice after intravenous injection of Met@Man-MPs pretreated with or without Bati (0.6 mg mL$^{-1}$) at the Met dosage of 10 mg kg$^{-1}$ every two days for six times, or/and intraperitoneal injection of Etan at a dosage of 5 mg kg$^{-1}$ every four days for four times indicated in (**a**). The tumor vessels were labeled with FITC-conjugated CD31 antibody (green) and nuclei were stained by DAPI (blue). The tumor margin was marked by a white dotted line. Images are representative of three biologically independent mice. Scale bars: 200 μm. **c** CD8+ T cells number in the interior or margin areas of tumor tissues of H22 tumor-bearing mice after treatment indicated in (**b**). Data are presented as mean ± s.d. ($n = 9$ fields for 3 mice; one-way ANOVA followed by Tukey's HSD post-hoc test). **d–f** The numbers of CD8+ T cells (**d**), CD8+CD69+ T cells, **e** and CD8+IFN-γ+ T cells (**f**) in tumor tissues of H22 tumor-bearing mice after treatment indicated in (**a**). Data are presented as mean ± s.d. ($n = 6$ mice per group; one-way ANOVA followed by Tukey's HSD post-hoc test). **g** Tumor volume of H22 tumor-bearing mice after treatment indicated in **a**. Data are presented as mean ± s.e.m. ($n = 6$ mice per group; two-way ANOVA followed by Bonferroni's multiple comparisons post-test). **h** Tumor weight of H22 tumor-bearing mice after treatment indicated in (**a**). Data are presented as mean ± s.d. ($n = 6$ mice per group; one-way ANOVA followed by Tukey's HSD post-hoc test). Source data are provided as a Source Data file.

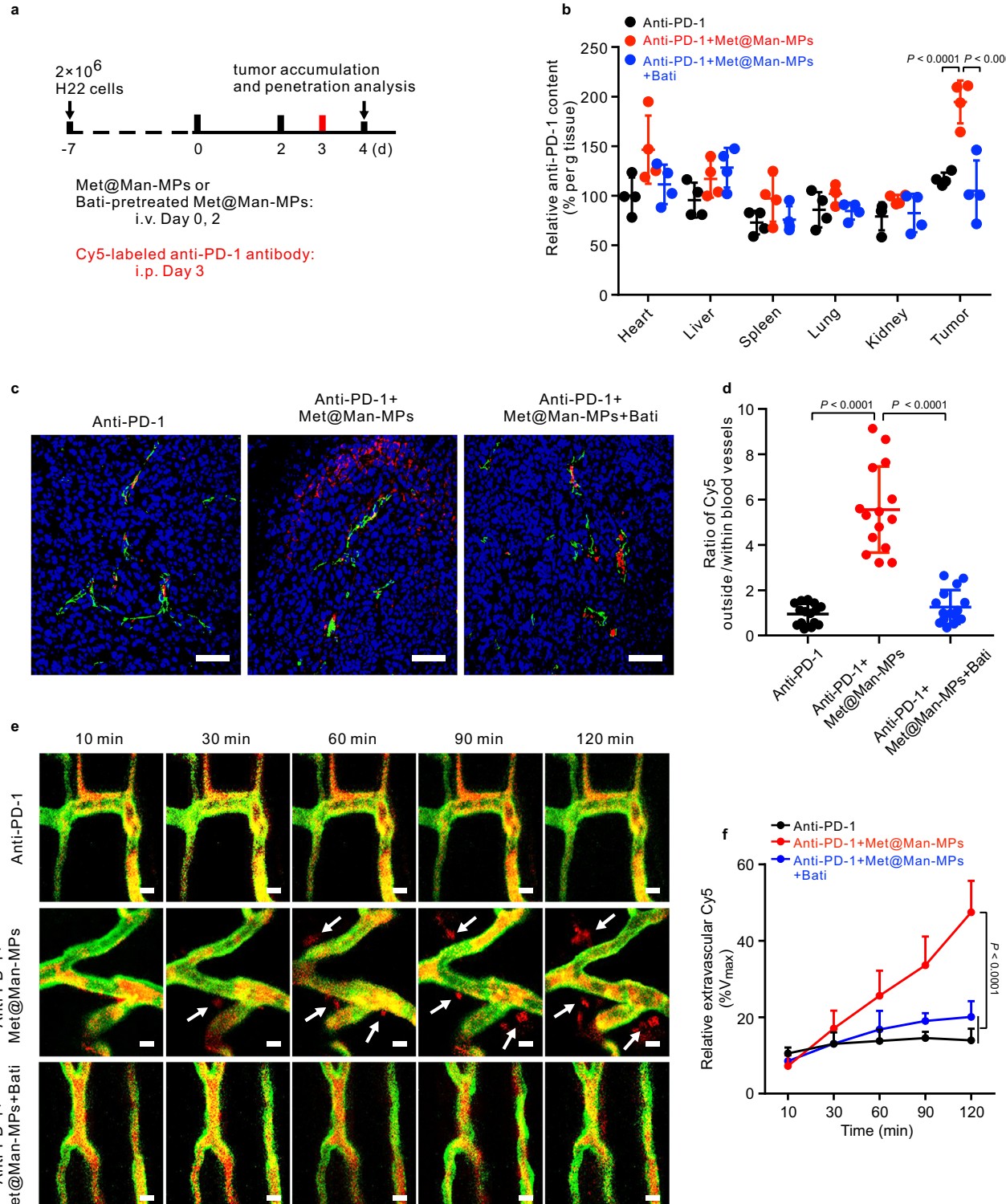

tumor accumulation and penetration of anti-PD-1 antibody induced by the collagen degradation of Man-MPs. A combination of Met@Man-MPs and anti-PD-1 antibodies generated the strongest synergistic anticancer effects with no significant toxicity (Supplementary Fig. 36), resulting in 60% of mice being tumor-free. Kaplan–Meier survival analysis showed that 100% of mice were still alive in the Met@Man-MPs and anti-PD-1 antibody-treated group after 70 d, better than any of the other groups. Importantly, compared with other groups, the CD8[+] effector memory T cells in spleens were significantly increased in Met@Man-MPs and anti-

PD-1 antibody-treated group (Fig. 8j). To further investigate the effects of co-administration of Met@Man-MPs and anti-PD-1 antibodies on the memory T-cell response in vivo, we used a tumor rechallenge model. When the mice with complete response (CR) in the Met@Man-MPs and anti-PD-1 antibody-treated group were again inoculated with H22 cells at 70 d after initial treatment, 100% of the mice rejected the tumor rechallenge up to 25 d, while continuous tumor growth was observed in naïve mice inoculated in parallel with the same number of cells (Fig. 8k). Meanwhile, 100% of tumor formation was observed in these mice with CR when

**Fig. 7 Enhanced tumor accumulation and penetration of anti-PD-1 antibody by Met@Man-MPs. a** Schematic schedule for tumor accumulation and penetration analysis of anti-PD-1 antibody in H22 tumor-bearing mice. **b** Relative anti-PD-1 antibody content in different tissues and tumors of H22 tumor-bearing mice after treatment with Cy5-labeled anti-PD-1 antibody (100 μg per mice) and Met@Man-MPs pretreated with or without Bati (0.6 mg mL$^{-1}$) at the Met dosage of 10 mg kg$^{-1}$ indicated in (**a**). Data are presented as mean ± s.d. ($n = 4$ mice per group; one-way ANOVA followed by Tukey's HSD post-hoc test). **c** Colocalization of Cy5-labeled anti-PD-1 antibody (red) with endothelial cells labeled with FITC-conjugated CD31 antibody (green) in tumor sections of H22 tumor-bearing mice after treatment with Cy5-labeled anti-PD-1 antibody (100 μg per mice) and Met@Man-MPs pretreated with or without Bati (0.6 mg mL$^{-1}$) at the Met dosage of 10 mg kg$^{-1}$ indicated in (**a**). The nuclei were stained by DAPI (blue). Images are representative of 3 biologically independent mice. Scale bars: 50 μm. **d** Ratio of Cy5 in tumor parenchyma to that within blood vessels as indicated in **c**. Data are presented as mean ± s.d. ($n = 15$ fields for 3 mice; one-way ANOVA followed by Tukey's HSD post-hoc test). **e** Time-lapse imaging of Cy5 extravasation and penetration into tumor tissues of H22 tumor-bearing mice after the mice were intravenously injected with Met@Man-MPs pretreated with or without Bati (0.6 mg mL$^{-1}$) at the Met dosage of 10 mg kg$^{-1}$ every other day for two times and then intravenously injected with Cy5-labeled anti-PD-1 antibody (100 μg per mice, red). The vasculature was labeled with fluorescein-labeled tomato lectin (green). The arrows indicated the extravascular anti-PD-1 antibody. Images are representative of two biologically independent mice. Scale bars: 10 μm. **f** Extravascular Cy5 fluorescence normalized by the maximum Cy5 fluorescence within blood vessels at 10 min post-injection (expressed as %Vmax) indicated in **e**. Data are presented as mean ± s.d. ($n = 10$ fields for 2 mice; two-way ANOVA followed by Bonferroni's multiple comparisons post-test). Source data are provided as a Source Data file.

inoculated with 4T1 cells (Fig. 8l). These results suggest that the combined treatment with Met@Man-MPs and anti-PD-1 antibody induces long-term antitumor effects by generating immunological memory. The excellent synergistic antitumor activity of the combination of Met@Man-MPs and anti-PD-1 antibody was further confirmed in 4T1 orthotopic tumor model with high malignancy. Anti-PD-1 antibody did not generate significant inhibition in tumor growth and lung metastasis of 4T1 tumor-bearing mice (Supplementary Fig. 37a–e). However, a combination of Man-MPs and anti-PD-1 antibodies significantly halted the development of the tumor mass (Supplementary Fig. 37b) and inhibited the lung metastasis (Supplementary Fig. 37c–e). Combination of Met@Man-MPs and anti-PD-1 antibody generated the strongest synergistic anticancer effects, achieving 75.0% tumor shrinkage (Supplementary Fig. 37b), 81.2% metastasis inhibition (Supplementary Fig. 37c, d) and longer survival time (Supplementary Fig. 37f).

The improved anticancer activity and tumor immune microenvironment induced by the combination of Met@Man-MPs and anti-PD-1 antibody were further confirmed in azoxymethane (AOM)/dextran sodium sulfate (DSS)-induced colitis-associated cancer (CAC) mouse model (Fig. 9a). Consistently, Met@Man-MPs significantly decreased the number of large tumors (Fig. 9b and Supplementary Fig. 38). The anticancer activity of anti-PD-1 antibody was significantly improved by Man-MPs (Fig. 9b and Supplementary Fig. 38). The combination of Met@Man-MPs and anti-PD-1 antibody generated the strongest synergistic inhibition in large tumors (Fig. 9b and Supplementary Fig. 38). H&E staining also confirmed the strongest anticancer effects of combination of Met@Man-MPs and anti-PD-1 antibody (Supplementary Fig. 39). Combination of Met@Man-MPs and anti-PD-1 antibody-treated mice lost less body weight (Supplementary Fig. 40). Meanwhile, tumor immune microenvironment analysis showed that Met@Man-MPs significantly increased the percentages of M1-like TAMs (Fig. 9c), CD8$^+$ T cells (Fig. 9e), activated CD8$^+$ T cells (Fig. 9f), and activated CD4$^+$ T cells (Fig. 9h), while decreased the percentages of M2-like TAMs (Fig. 9d), MDSCs (Fig. 9i) and Tregs (Fig. 9j). However, the combination of Met@Man-MPs and anti-PD-1 antibodies exhibited the strongest capacity to reshape tumor immune microenvironment (Fig. 9c–j).

To determine whether the combination of Met@Man-MPs and anti-PD-1 antibody could promote the antitumor immunity in situ (Supplementary Fig. 41), we employed an organotypic ex vivo slice culture model of fresh human tumors. The organotypic slices from one liver cancer patient-derived tumor were incubated with peripheral blood mononuclear cells (PBMCs) from the same patient in the presence of Met@Man-MPs derived from THP-1-originated macrophages, anti-PD-1 antibody or both Met@Man-MPs and anti-PD-1 antibody.

Consistently, Met@Man-MPs significantly induced apoptosis in tumor cells (Fig. 10a). Meanwhile, Met@Man-MPs markedly reshaped the tumor immune microenvironment, as evidenced by the increased M1-like TAM percentages (Fig. 10b), the decreased M2-like TAM percentages (Fig. 10c), and the enhanced infiltration and activation of CD8$^+$ T cells (Fig. 10d–g) in tumor tissues. However, the strongest apoptosis induction in tumor cells (Fig. 10a) and the strongest CD8$^+$ T cell activation (Fig. 10d–f) were observed in the Met@Man-MPs and anti-PD-1-treated group, which was further confirmed in organotypic slices from another liver cancer patient-derived tumor (Supplementary Fig. 42). These results strongly support the notion that Met@-Man-MPs efficiently reshape the tumor microenvironment and improve the anticancer activity of anti-PD-1 antibody.

## Discussion

In this study, we developed Man-modified macrophage-derived MPs loading Met (Met@Man-MPs) to boost the anticancer activity of the anti-PD-1 antibody. Met@Man-MPs were highly accumulated in M2-like TAMs compared with other immune cells and tumor cells in tumor tissues and other tissue-resident macrophages, resulting in the repolarization of M2-like TAMs into M1 phenotype. The Met@Man-MPs-reset macrophages not only directly killed and phagocytosed tumor cells to inhibit tumor growth, but also probably generated tumor antigen to increase the tumor recognition capability of cytotoxic T cells, potentiating anti-PD-1 treatment efficacy. Meanwhile, Met@Man-MPs-reset macrophages efficiently reconstructed antitumor immune microenvironment to improve anti-PD-1 therapy by decreasing the numbers of Tregs and MDSCs and increasing the recruitment of CD8$^+$ T cells in tumor tissues, as evidenced by the fact that macrophage depletion using clodronates liposomes significantly abrogated the improved tumor immune microenvironment induced by Met@Man-MPs. The recruitment of CD8$^+$ T cells into tumor tissues might be due to the secreted chemokines and cytokines by Met@Man-MPs-reset macrophages, such as TNF-α, since TNF-α expression and secretion was significantly upregulated in Met@Man-MPs-treated M2-like macrophages, and Etan, an inhibitor of TNF-α, significantly decreased the recruitment of CD8$^+$ T cells by Met@Man-MPs-treated M2-like macrophages by transwell migration assay. More importantly, intraperitoneal injection of Etan markedly abrogated the enhanced number of CD8$^+$ T cells in tumor tissues induced by Met@Man-MPs in H22 tumor-bearing mice. However, whether other chemokines and cytokines were involved in the Met@Man-MPs-induced recruitment of CD8$^+$ T cells remains to be elucidated.

Besides enhanced tumor immunogenicity, reshaped tumor immunosuppressive microenvironment, and increased CD8$^+$ T

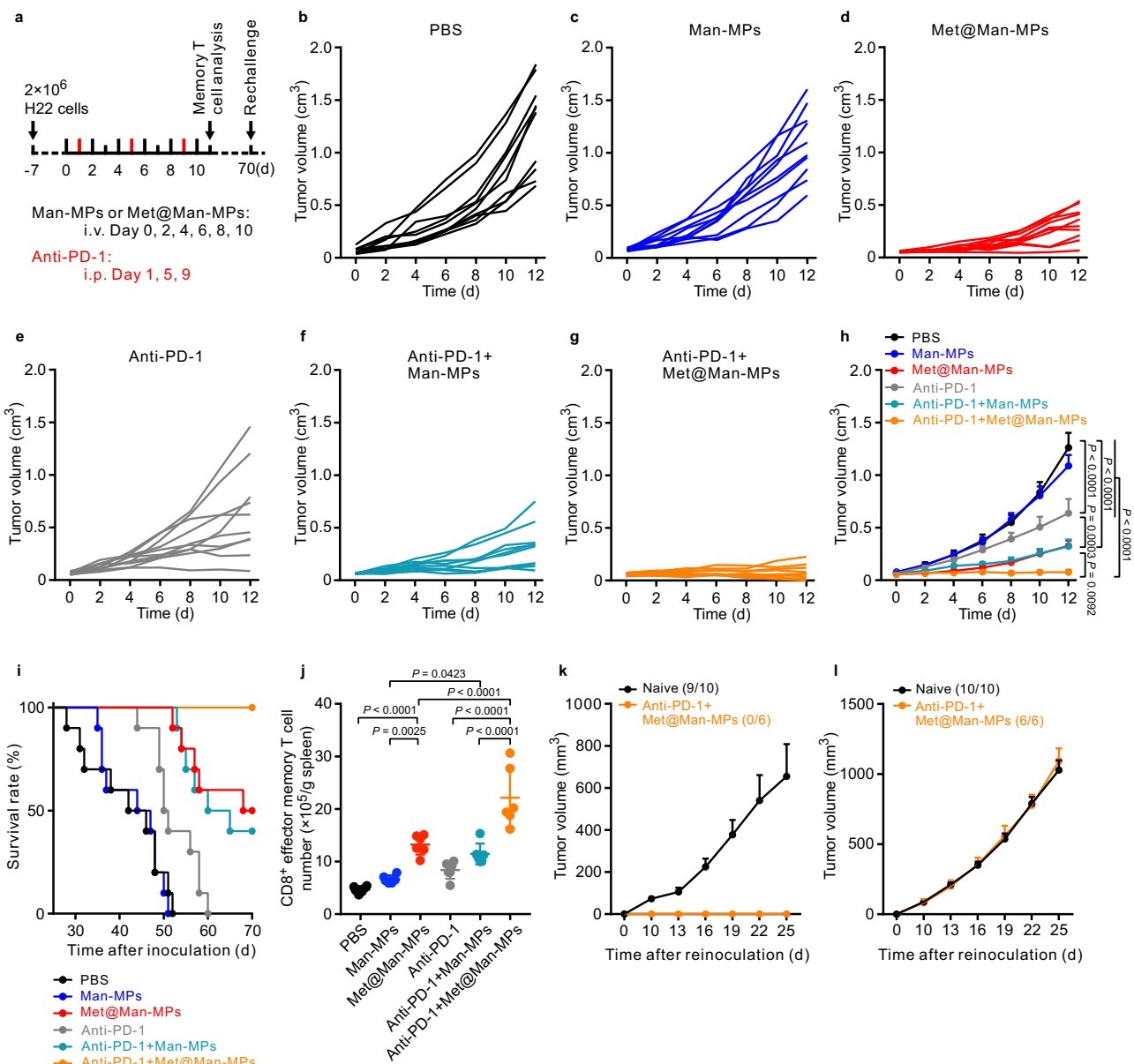

**Fig. 8 Anticancer activity of combination of anti-PD-1 antibody and Met@Man-MPs in H22 tumor-bearing mice. a** Schematic schedule for an anticancer experiment in H22 tumor-bearing mice. **b–g** Individual tumor growth curves of H22 tumor-bearing mice after treatment with PBS (**b**), Man-MPs (**c**), Met@Man-MPs (**d**), anti-PD-1 antibody (**e**), a combination of Man-MPs and anti-PD-1 antibody (**f**), or combination of Met@Man-MPs and anti-PD-1 antibody (**g**) at the anti-PD-1 antibody dosage of 100 μg per mouse and Met dosage of 10 mg kg$^{-1}$ indicated in (**a**). **h** Average tumor growth curves of H22 tumor-bearing mice indicated in **a**. Data are presented as mean ± s.e.m. ($n = 10$ mice per group; two-way ANOVA followed by Bonferroni's multiple comparisons post-test). **i** Kaplan–Meier survival plot of H22 tumor-bearing mice after treatment indicated in (**a**) ($n = 10$ mice per group). **j** The numbers of CD8$^+$ effector memory T cells (CD3$^+$CD8$^+$CD44$^+$CD62L$^-$T cells) in spleens of H22 tumor-bearing mice after treatment indicated in **a**. Data are presented as mean ± s.d. ($n = 6$ mice per group; one-way ANOVA followed by Tukey's HSD post-hoc test). **k, l** Tumor growth curve after rechallenge with H22 cells ($3 \times 10^6$ cells, **k**) and 4T1 cells ($1 \times 10^6$ cells, **l**) in naïve mice or combination of anti-PD-1 antibody and Met@Man-MPs-treated mice indicated in (**a**). The ratio of mice with tumorgenesis is indicated in the brackets. Data are presented as mean ± s.e.m. ($n = 10$ for naïve mice, $n = 6$ for anti-PD-1 antibody- and Met@Man-MPs-treated mice). Source data are provided as a Source Data file.

cell recruitment, the infiltration of CD8$^+$ T cells into tumor interiors and the enhanced tumor accumulation and penetration of PD-1 antibody are also important for the improved anti-PD-1 therapy[6,9,10]. Man-MPs overexpressed MMPs, such as MMP9 and MMP14 proteins, leading to the efficient degradation of tumor collagen. The Man-MPs-triggered degradation of tumor collagen favored the infiltration of CD8$^+$ T cells into tumor interiors and tumor accumulation and penetration of anti-PD-1 antibody since pretreatment with Bati, an MMP inhibitor, markedly abrogated these effects induced by Met@Man-MPs.

Importantly, H&E staining analysis showed that the Man-MPs-triggered degradation of tumor collagen did not induce tumor metastasis in H22 tumor-bearing mice, showing good biosafety. However, which MMP plays an important role in Man-MPs-triggered degradation of tumor collagen needs further clarification. Meanwhile, Bati, like its derivative marimastat having a collagen-mimicking hydroxamate structure, can bind to off-target metalloproteinases that are not MMPs[55]. Besides MMPs, whether other metalloproteinases are involved in the biological function of Met@Man-MPs remains to be elucidated.

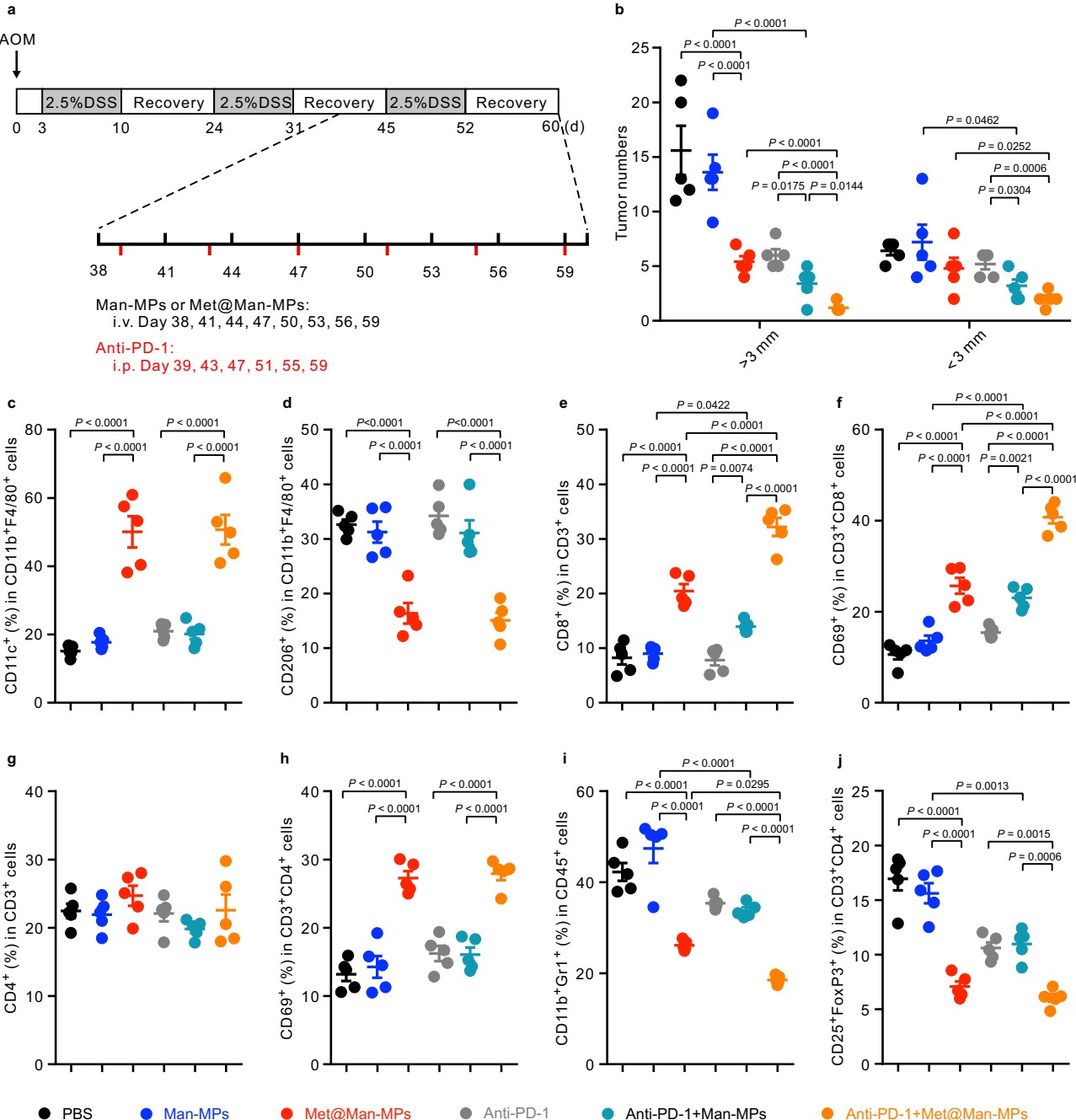

**Fig. 9 Anticancer activity and improved tumor immune microenvironment of a combination of anti-PD-1 antibody and Met@Man-MPs in AOM/DSS-induced CAC mice. a** Schematic schedule for anticancer experiment in AOM/DSS-induced CAC mice. **b** Numbers of tumors with size>3 mm and <3 mm in CAC mice after treatment with PBS, Man-MPs, Met@Man-MPs, anti-PD-1 antibody, a combination of Man-MPs and anti-PD-1 antibody, or combination of Met@Man-MPs and anti-PD-1 antibody at the anti-PD-1 antibody dosage of 100 µg per mouse and Met dosage of 10 mg kg$^{-1}$ indicated in (**a**). Data are presented as mean ± s.d. ($n = 5$ mice per group; one-way ANOVA followed by Tukey's HSD post-hoc test). **c–j** Percentages of M1-like TAMs (**c**), M2-like TAMs (**d**), CD8$^+$ T cells (**e**), CD8$^+$CD69$^+$ T cells (**f**), CD4$^+$ T cells (**g**), CD4$^+$CD69$^+$ T cells (**h**), MDSCs (**i**) and Tregs (**j**) in colon tumors of CAC mice after treatment with PBS, Man-MPs, Met@Man-MPs, anti-PD-1 antibody, a combination of Man-MPs and anti-PD-1 antibody, or combination of Met@Man-MPs and anti-PD-1 antibody at the anti-PD-1 antibody dosage of 100 µg per mouse and Met dosage of 10 mg kg$^{-1}$ indicated in (**a**). Data are presented as mean ± s.d. ($n = 5$ mice per group; one-way ANOVA followed by Tukey's HSD post-hoc test). Source data are provided as a Source Data file.

In view of the above facts, Met@Man-MP efficiently integrated all features to improve the therapeutic efficacy of anti-PD1 antibody, generating remarkable anticancer activity after coadministration in different tumor models, including H22 tumor-bearing mice, 4T-1 tumor-bearing mice, AOM/DSS-induced CAC mice, and even an organotypic ex vivo slice culture model of fresh human tumors. Here, Met@Man-MPs derived from RAW264.7 cells, BMDMs, THP-1-originated macrophages, and human MDMs showed similar repolarization activity of M2-like macrophages to M1 phenotype, and expressed MMP, showing that the features of MPs derived from different macrophages are universal. To maximize the treatment efficacy of Met@Man-MPs, several areas should be further explored. Particle size critically affects the pharmacokinetics and tissue biodistribution.

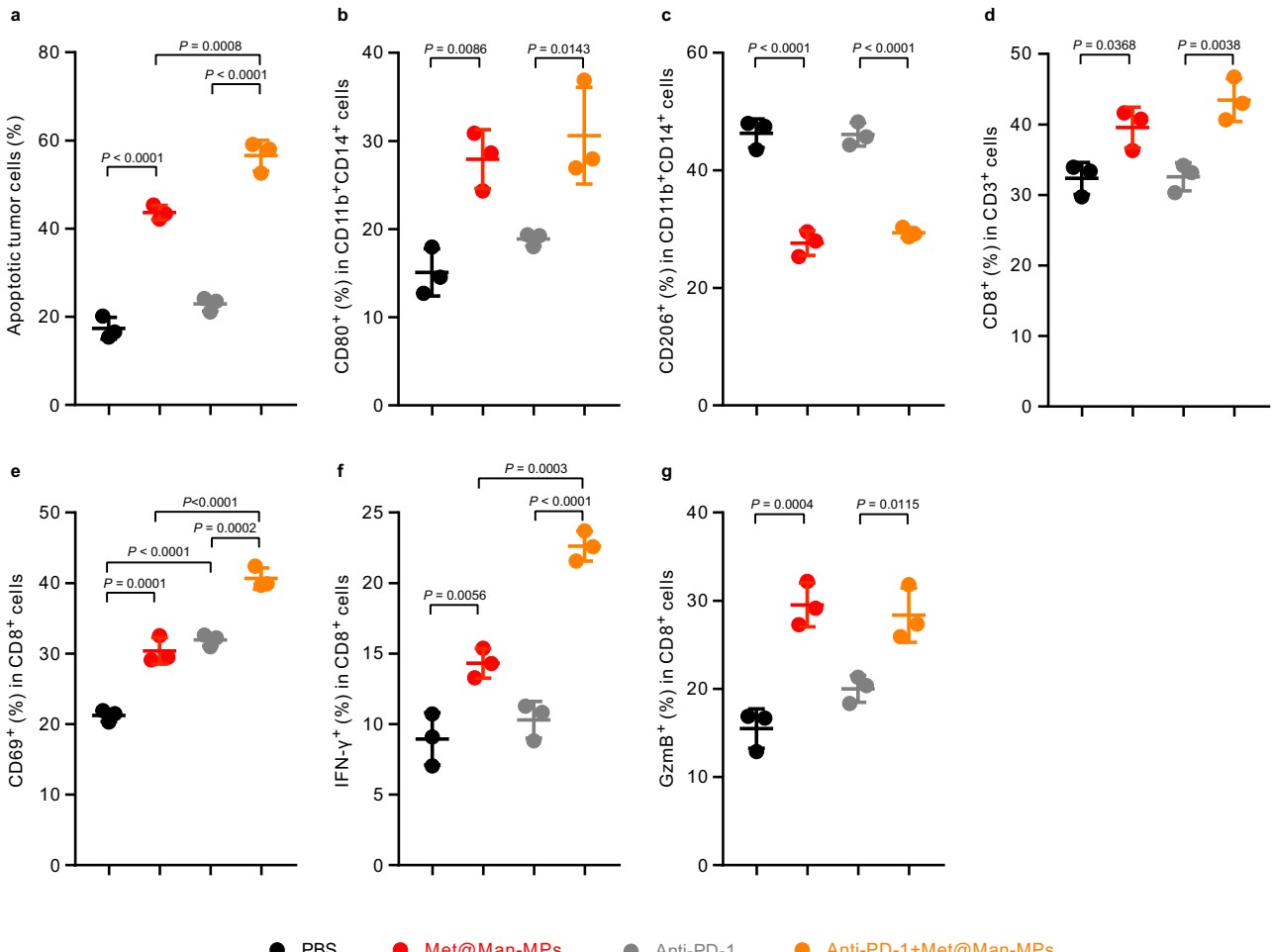

**Fig. 10 Apoptosis and improved tumor immune microenvironment induced by combination of anti-PD-1 antibody and Met@Man-MPs in organotypic slices from one liver cancer patient-derived tumor. a** The ratio of apoptotic tumor cells (CD45[−] cells) after the tumor slices from patient-derived tumor were treated with anti-human PD-1 antibody, Met@Man-MPs derived from THP-1-orignated macrophages or Met@Man-MPs plus anti-human PD-1 antibody at the concentration of anti-PD-1 antibody and Met of 20 and 40 μg mL[−1] in the presence of PBMCs for 36 h, respectively. Data are presented as mean ± s.d. ($n = 3$ biologically independent samples; one-way ANOVA followed by Tukey's HSD post-hoc test). (**b–g**) Percentages of M1-like TAMs (**b**), M2-like TAMs (**c**), CD8[+] T (**d**), CD8[+]CD69[+] T (**e**), CD8[+]IFN-γ[+] T (**f**), and CD8[+]GzmB[+] T cells (**g**) in the tumor slices from patient-derived tumor after treatment with anti-human PD-1 antibody, Met@Man-MPs derived from THP-1-orignated macrophages or Met@Man-MPs plus anti-human PD-1 antibody at the concentration of anti-PD-1 antibody and Met of 20 and 40 μg mL[−1] in the presence of PBMCs for 36 h, respectively. Data are presented as mean ± s. d. ($n = 3$ biologically independent samples; one-way ANOVA followed by Tukey's HSD post-hoc test). Source data are provided as a Source Data file.

Developing Man-MPs with a narrower size distribution would be ideal for the intravenous delivery of Met; Recently, M1-like macrophage-derived exosomes were reported to repolarize M2-like TAMs to M1 phenotype[56], and be uptaken by DCs and macrophages to induce the release of a pool of Th1 cytokines[57]. Thus, the development of Man-modified MPs derived from M1-like macrophages to load Met might exert better anticancer activity and improved tumor microenvironment to potentiate the therapeutic effects of anti-PD-1 antibody; Optimization of Met loading content and the dosing schedule is also required.

Despite the encouraging efficacy and satisfactory safety of a combination of Met@Man-MPs and anti-PD-1 antibody in immunocompetent mouse models, great care has been taken to translate this treatment platform from bench to human clinical trials. First, the macrophage resource for Man-MP fabrication would be a critical issue. For clinical application, it would be better to use human MDMs rather than immortalized cells as the donor cells to prepare Met@Man-MPs because cell lines would not have the necessary "self" characteristics to avoid recognition

by the immune system[58]. However, the availability of human MDMs and the development of a robust, scalable process would be a challenge. MPs derived from immortalized tumor cell lines have been applied to cancer patients in clinical settings[59]. Addressing the potential risks of the immune response induced by allogeneic cell-derived MPs needs to be further elucidated. Second, quality control standards for MP constitution and content, and drug encapsulation efficiency and loading content in the GMP production department are essential.

In summary, the data in this study show that Met@Man-MPs efficiently target to M2-like TAMs to repolarize into M1-like phenotype, resulting in the recruitment of CD8[+] T cells into tumor tissues and the ameliorated tumor immunosuppressive microenvironment. In addition, the collagen-degrading capacity of Man-MPs contributes to the infiltration of CD8[+] T cells into tumor interiors and the enhanced tumor accumulation and penetration of anti-PD-1 antibody. With these unique features, Met@Man-MP efficiently improve the anticancer activity of anti-PD1 antibody therapy, generating long-term memory immunity. These findings

reveal that Met@Man-MPs may add to the therapeutic tools for tumor resistance to current anti-PD-1 therapy.

## Methods

**Materials**. RPMI 1640 medium, Dulbecco's Modified Eagle's Medium (DMEM) and FBS was purchased from Gibco BRL/Life Technologies (Grand Island, NY, USA). 2-distearoyl-sn-glycero-3-phosphoethanolamine-$N$-[carboxy(polyethylene glycol) -2000] sodium salt (DSPE-PEG-COOH) was purchased from Ponsure (Shanghai, China). IL-4 and IFN-γ were provided by PeproTech (Rocky Hill, NJ, USA). D-Mannosamine hydrochloride was purchased from Aladdin (Shanghai, China). PKH26, PKH67, LPS, and AOM were provided by Sigma-Aldrich (St Louis, MO, USA). Metformin hydrochloride was obtained from J&K China Chemical Ltd. (Beijing, China). DSS was purchased from MP Biomedicals (Santa Ana, CA, USA). Antibodies used for flow cytometry analysis were purchased from BioLegend (San Diego, CA, USA). All other reagents were purchased from Sinopharm Chemical Reagent Co. Ltd. (Shanghai, China) and were of analytical grade.

**Cell culture and animals**. RAW264.7, H22, 4T1, THP-1, and HepG2 cells were purchased from Type Culture Collection of the Chinese Academy of Sciences (Shanghai, China). Murine dendritic cell line DC2.4 (H-2b) was kindly provided by Dr. Haifang Yin (Tianjin Medical University, Tianjin, China). RAW264.7, DC2.4, and HepG2 cells were cultured in DMEM medium, and H22, THP-1, and 4T1 cells were maintained in RPMI 1640 medium containing 10% FBS, 100 U mL$^{-1}$ penicillin, and 100 μg mL$^{-1}$ streptomycin at 37 °C in a 5% CO$_2$ incubator. THP-1-derived macrophages were obtained by stimulating THP-1 cells with 100 ng mL$^{-1}$ Phorbol-12-myristate-13-acetate (PMA) for 24 h. Murine BMDMs were obtained by isolating bone marrow cells from femurs and tibias of 6–8 weeks old BALB/c male mice and then culturing bone marrow cells in the RPMI 1640 complete growth medium containing 20 ng mL$^{-1}$ recombinant mouse macrophage colony-stimulating factor (M-CSF) for 5 days. Murine BMDCs were obtained by culturing bone marrow cells in RPMI 1640 complete growth medium containing 10 ng mL$^{-1}$ recombinant mouse granulocyte-macrophage colony-stimulating factor (GM-CSF) and 5 ng mL$^{-1}$ IL-4. Human MDMs were obtained by isolating PBMCs from healthy volunteers and then culturing them in the RPMI 1640 complete growth medium containing 20 ng mL$^{-1}$ recombinant human M-CSF for 7 days. The study was approved by the Clinical Trial Ethics Committee of Huazhong University of Science and Technology and the volunteers provided informed consent. M1-like macrophages were obtained by stimulating RAW264.7 cells or BMDMs (M0 macrophages) with 100 ng mL$^{-1}$ LPS and 20 ng mL$^{-1}$ IFN-γ for 24 h, and M2-like macrophages were obtained by stimulating RAW264.7 cells, BMDMs, THP-1-derived macrophages or human MDMs with 20 ng mL$^{-1}$ IL-4 for 24 h.

BALB/c mice (male and female, 18–20 g) were purchased from Beijing Vital River Laboratory Animal Technology Co., Ltd. (Beijing, China). Mice were housed in an animal facility under constant environmental conditions (room temperature, 21 ± 1 °C; relative humidity, 40−70%, and a 12-h light-dark cycle). All mice had access to food and water ad libitum. H22 tumor-bearing mice were constructed by subcutaneous injection of H22 cells ($2 \times 10^6$ cells per mouse) into the flanks of male BALB/c mice. The orthotopic 4T1 breast cancer model was constructed by inoculating 4T1 cells ($3 \times 10^5$ cells per mouse) into the right forth breast fat pad of BALB/c female mice. All animal experiments were performed under the guidance and approved by the Institutional Animal Care and Use Committee at Tongji Medical College, Huazhong University of Science and Technology (Wuhan, China).

**Preparation and characterization of Met@Man-MPs**. To collect Met@Man-MPs, DSPE-PEG-Man was first synthesized by conjugating DSPE-PEG-COOH with D-mannosamine hydrochloride via EDC/NHS coupling reaction. RAW264.7 cells were incubated with DSPE-PEG-Man (25 μg mL$^{-1}$) for 3 days. The cells incubated with or without DSPE-PEG-Man were then treated with 2 mg mL$^{-1}$ Met, followed by ultraviolet irradiation (UVB, 300 J m$^{-2}$) for 1 h. At 12 h after treatment, the supernatants were centrifuged at $600 \times g$ for 10 min to get rid of cells and cell debris. The supernatants were further centrifuged at $18,000 \times g$ for 60 min to collect Met@Man-MPs and Met@MPs, respectively. The pellets were washed three times and resuspended in sterile PBS for further use. The blank Man-MPs and MPs were collected according to the same protocol, except that no Met was added. The concentration of Met in MPs or Man-MPs was measured by lysing these MPs with 1% SDS and extracting Met with methyl alcohol, followed by quantification by an HPLC system (Agilent 1100, USA) under the following chromatographic conditions: column, C18 column ($5 \times 250$ mm, particle size 5 μm); mobile phase, methyl alcohol-15 mM KH$_2$PO$_4$ (60:40, v/v); flow rate, 1.0 mL min$^{-1}$; detection wavelength, 233 nm. The hydrodynamic diameter and zeta potential of Met@Man-MPs were determined by Zetasizer Nano ZS 90 (Malvern Instruments Ltd., Worcestershire, UK). Their morphology was observed by TEM.

**Cellular uptake**. M0, M1, and M2-like macrophages, DC2.4, and H22 cells were treated with PKH67-labeled MPs or Man-MPs at the concentration of 10 μg protein mL$^{-1}$ for 4 h. The cells were then rinsed with PBS three times. The intracellular PKH67 fluorescence was detected using FC500 flow cytometry (Beckman Coulter, Fullerton, CA, USA) and analyzed using CXP cytometer software

(Beckman Coulter). Furthermore, M0, M1, and M2-like macrophages, HepG2 and H22 cells were treated with free Met, Met@MPs or Met@Man-MPs at Met concentration of 40 μg mL$^{-1}$. After 4-h incubation, the intracellular Met content was detected by HPLC.

**Biodistribution analysis**. When the tumor volume of H22 tumor-bearing mice reached around 200 mm$^3$, the mice were intravenously administered with Cy5-labeled MPs or Man-MPs at the dosage of 15 mg protein kg$^{-1}$, or free Met, Met@MPs, Met@Man-MPs at the Met dosage of 10 mg kg$^{-1}$ or free Met at a high dosage of 100 mg kg$^{-1}$. At 24 h after administration, the mice were executed and the major organs (heart, liver, spleen, lung, and kidney) and tumors were removed. The NIRF images of organs and tumors were obtained by a Caliper IVIS Lumina II in vivo imaging system (PerkinElmer, Waltham, MA, USA). For measurement of Met content, the samples were weighed and homogenized in PBS. Met was extracted by incubating 100 μL of homogenates in 100 μL of methyl alcohol. The Met contents in the lysates were measured using HPLC.

**Cellular uptake assay in vivo**. GFP-expressing H22 tumor-bearing mice were intravenously injected with PKH26-labeled MPs or Man-MPs at the dosage of 15 mg protein kg$^{-1}$. At 24 h after injection, the mice were sacrificed, and the livers, spleens, lungs, and tumors were collected. The tissues were cut into small pieces and incubated in RPMI 1640 medium containing 0.8 mg mL$^{-1}$ collagenase type I and 5 μg mL$^{-1}$ DNase I at 37 °C for 30 min. The single-cell suspensions were harvested by washing twice with PBS and filtering twice through a 40 μm cell strainer. Tumor-infiltrating lymphocytes (TILs) were isolated by Ficoll-Paque PLUS density gradient media (GE Healthcare, MA, USA). For PKH26 fluorescence analysis in TAMs, Kupffer cells, splenic macrophages, and pulmonary macrophages, the cells were stained with fluorescence-labeled anti-CD11b (Biolegend, cat. No 101227, clone M1/70, 1/80 dilution) and F4/80 (Biolegend, cat. No 123137, clone BM8, 1/100 dilution). For M1/M2-like TAMs analysis, the cells were stained with fluorescence-labeled anti-CD11b, F4/80, CD206 (Biolegend, cat. No 141707, clone C068C2, 1/50 dilution) and CD11c (Biolegend, cat. No 117317, clone N418, 1/80 dilution). For PKH26 fluorescence analysis in DCs, the cells were stained with fluorescence-labeled anti-CD45 (Biolegend, cat. No 103112, clone 30−F11, 1/100 dilution), F4/80 and CD11c. For PKH26 fluorescence analysis in CD4$^+$ and CD8$^+$ T cells, the TILs were stained with fluorescence-labeled anti-CD45, CD3 (Biolegend, cat. No 100220, clone 17A2, 1/100 dilution), CD4 (Biolegend, cat. No 100539, clone RM4-5, 1/80 dilution) or CD8 (Biolegend, cat. No 100737, clone 53-6.7, 1/20 dilution). For PKH26 fluorescence analysis in MDSCs, the TILs were stained with fluorescence-labeled anti-CD45, CD11b, and Ly-6G/Ly-6C (Gr-1) (Biolegend, cat. No 108433, clone RB6-8C5, 1/20 dilution). For PKH26 fluorescence analysis in Tregs, TILs cells were treated with transcription factor buffer set (BD pharmingen, cat. No 562574) after staining with fluorescence-labeled anti-CD3, CD4, and CD25 (Biolegend, cat. No 102011, clone PC61, 1/100 dilution) and then re-stained with fluorescence-labeled anti-FoxP3 (Biolegend, cat. No 126419, clone MF-14, 1/50 dilution). All antibodies were diluted according to the instructions and incubated with the cells for 30 min at room temperature. The PKH 26 fluorescence in these cells was determined by flow cytometry (CytoFLEX S, Beckman Coulter, Fullerton, CA, USA) and analyzed by using CytExpert (Beckman Coulter) software.

**M2 macrophage repolarization**. M2-like macrophages were treated with PBS, MPs, Man-MPs, free Met, Met@MPs, Met@Man-MPs at the Met concentration of 20 μg mL$^{-1}$, or free Met at a high concentration of 200 μg mL$^{-1}$. After 24-h treatment, the cells were harvested and the expression of M1-related markers (*iNOS*, *TNF-α*, CD80, and CD86) and M2-related markers (*Arg1*, *Mrc1*, *Mgl1*, *CD163*, and CD206) were evaluated by real-time reverse transcription-polymerase chain reactions (RT-PCR) or flow cytometry. The used primer sequences for RT-PCR are as follows: mouse *GAPDH* (F: 5′-GTTCCTACCCCCAATGTGTCC-3′, R: 5′-TAGCCCAAGATGCCCTTCAGT-3′); mouse *iNOS* (F: 5′-GATGTTGAACTA TGTCCTATCTCC-3′, R: 5′-GAACACCACTTTCACCAAGAC-3′); mouse *TNF-α* (F: 5′-CCCACGTCGTAGCAAACCAC-3′, R: 5′-GCAGCCTTGTCCCTTGAA GA-3′); mouse *Arg1* (F: 5′-CAAGACAGGGGCTCCTTTCAG-3′, R: 5′-TGGCTT ATGGTTACCCTCCC-3′); mouse *Mrc1* (F: 5′-ATGGGCAACATCGAGCAG AA-3′, R: 5′-AAACCAATGCAACCCAGTGC-3′); mouse *Mgl1* (F: 5′-AGAA AACCCAAGAGCCTGGT-3′, R: 5′-GAGGCCCAGGGAGAACAG-3′); human *GAPDH* (F: 5′-GGAGCGAGATCCCTCCAAAAT-3′, R: 5′-GGCTGTTGTCA TACTTCTCATGG-3′); human *TNF-α* (F: 5′-GAGGCCAAGCCCTGGTATG-3′, R: 5′- CGGGCCGATTGATCTCAGC-3′); human *iNOS* (F: 5′-AGGGACAAGCC TACCCCCTC-3′, R: 5′-CTCATCTCCCGTCAGTTGGT-3′); human *CD80* (F: 5′-GGGCACATACGAGTGTTTGT-3′, R: 5′-TCAGCTTTGACTGATAACGTC AC-3′); human *CD86* (F: 5′-CTGCTCATCTATACACGGTTACC-3′, R: 5′-GG AAACGTCGTACAGTTCTGTG-3′); human *Mrc1* (F: 5′-GGGTTGCTATCAC TCTCTATGC-3′, R: 5′-TTTCTTGTCTGTTGCCGTAGTT-3′); human *CD163* (F: 5′-TTTGTCAACTTGAGTGCCCTTCAC-3′, R: 5′-TCCCGCTACACTTGTTT TCAC-3′).

**Cytotoxicity of repolarized M2-like macrophages against tumor cells**. M2-like macrophages were treated with PBS, MPs, Man-MPs, free Met, Met@MPs,

Met@Man-MPs at the Met concentration of 20 μg mL$^{-1}$, or free Met at a high concentration of 200 μg mL$^{-1}$ for 24 h. The supernatants were replaced with fresh media and cultured for 6 h to obtain conditional media. The conditional media were used to treat H22, 4T1, and HepG2 cells for 24 h. The cell viability was determined by CCK-8 assay (Dojindo, Kumamoto, Kyushu, Japan).

**Phagocytosis of tumor cells by repolarized M2-like macrophages.** M2-like macrophages were treated with PBS, MPs, Man-MPs, free Met, Met@MPs, Met@Man-MPs at the Met concentration of 20 μg mL$^{-1}$, or free Met at high concentration of 200 μg mL$^{-1}$ for 24 h. The treated M2-like macrophages labeled with CFSE were co-cultured with DiD-labeled tumor cells (H22 or 4T1 cells) at a ratio of 1:1 for 3 h at 37 °C. The phagocytosis of tumor cells by macrophages was detected as the ratio of the numbers of CFSE$^+$DiD$^+$ cells to CFSE$^+$ cells by flow cytometry.

**Macrophages and tumor cells co-transplantation.** M2-like macrophages were treated with PBS, MPs, Man-MPs, free Met, Met@MPs, Met@Man-MPs at the Met concentration of 20 μg mL$^{-1}$, or free Met at a high concentration of 200 μg mL$^{-1}$ for 24 h. The repolarized macrophages and H22 cells were co-implanted subcutaneously into the flanks of BALB/c mice at a ratio of 2:1 ($8 \times 10^5$ tumor cells per mouse). Palpable tumors were measured by calipers, and tumor volume was calculated according to the formula: volume = (length × width$^2$)/2. The mice were sacrificed, and then tumors were collected and weighed after 23 days.

**In vivo antitumor effects.** When tumor volume of H22 or 4T1 tumor-bearing mice reached around 50 mm$^3$, the mice were administrated intravenously with PBS, MPs, Man-MPs, free Met, Met@MPs, Met@Man-MPs derived from RAW264.7 cells at the Met dosage of 10 mg kg$^{-1}$, or high dosage of Met at 100 mg kg$^{-1}$ every 2 days for six times in the presence or absence of intraperitoneal injection of anti-PD-1 antibody at the dosage of 100 μg per mouse every 4 days for three times. The size of the tumor and the body weight of the mice were measured every day. The part of the mice was sacrificed, and the organs (heart, liver, spleen, lung, and kidney) and tumors were removed. The lungs of 4T1 tumor-bearing mice were fixed in Bouin's solution (Solarbio, Beijing, China) overnight at room temperature, and the metastatic tumor nodules in the lungs were counted. The organs were fixed with 4% paraformaldehyde, sectioned, and stained with H&E. The tumors were weighed, fixed with 4% paraformaldehyde, sectioned, and stained using a TUNEL assay kit (Roche, Mannheim, Germany) according to the manufacturer's protocol. The rest of the mice were kept for long-term tumor inhibition and survival observation.

For evaluating anticancer activity in AOM/DSS-induced CAC, C57BL/6 male mice (8-week old) were intraperitoneally injected with 10 mg/kg AOM. Three days after AOM injection, the mice were given 3 cycles of 2.5% DSS in drinking water for 7 days followed by drinking regular water for 14 days. On day 38, the mice were intravenously injected with PBS, Man-MPs, Met@Man-MPs derived from RAW264.7 cells at the Met dosage of 10 mg kg$^{-1}$ every 3 days for eight times in the presence or absence of intraperitoneal injection of anti-PD-1 antibody at the dosage of 100 μg per mouse every 4 days for six times. Body weight was monitored every 3 days. The animals were sacrificed on day 60, and colons were collected for further analysis. The tumor numbers in colons were counted and stained with H&E.

**Tumor immune microenvironment analysis.** After sacrificing the tumor-bearing mice, H22, 4T1 tumors, and colon tumors of CAC mice were dissected, cut into pieces and then incubated with RPMI 1640 media containing 0.8 mg mL$^{-1}$ collagenase and 5 μg mL$^{-1}$ DNase I for 60 min at 37 °C. The homogenates were washed with PBS and passed through a 40 μm nylon mesh to acquire single-cell suspensions. For M1/M2-like macrophages analysis, cells were stained with fluorescence-labeled anti-CD11b (Biolegend, cat. no 101206, clone M1/70, 1/200 dilution), F4/80 (Biolegend, cat. No 123110, clone BM8, 1/20 dilution), CD206 (Biolegend, cat. No 141707, clone C068C2, 1/50 dilution), CD11c (Biolegend, cat. No 117317, clone N418, 1/80 dilution). For lymphocyte analysis, TILs were isolated by Ficoll-Paque PLUS density gradient media (GE Healthcare). For surface marker, the cells were stained with fluorescence-labeled anti-CD3 (Biolegend, cat. No 100220, clone 17A2, 1/100 dilution; cat. No 100218, clone 17A2, 1/20 dilution), CD4 (Biolegend, cat. No 100406, clone GK1.5, 1/200 dilution), CD8 (Biolegend, cat. No 100707, clone 53-6.7, 1/100 dilution; cat. No 100722, clone 53-6.7, 1/80 dilution), CD69 (Biolegend, cat. No 104513, clone H1.2F3, 1/20 dilution), CD25 (Biolegend, cat. No 102011, clone PC61, 1/100 dilution), Ly-6G/Ly-6C (Gr-1) (Biolegend, cat. No 108433, clone RB6-8C5, 1/20 dilution). For intracellular cytokine staining, cells were treated with Fix/Perm solution (Biolegend, cat. No 420801) followed by re-staining with IFN-γ (Biolegend, cat. No 505809, clone XMG1.2, 1/20 dilution). For transcription factor staining, cells were treated with a transcription factor buffer set (BD pharmingen, cat. No562574) after surface staining and re-stained with anti-FoxP3 (Biolegend, cat. No 126419, clone MF-14, 1/50 dilution). Memory T cells in spleens were analyzed by staining with fluorescence-labeled anti-CD3 (Biolegend, cat. No100220, clone 17A2, 1/100 dilution; cat. No100236, clone 17A2, 1/50 dilution), CD8 (Biolegend, cat. No100737, clone 53-6.7, 1/20 dilution; cat. No100722, clone 53-6.7, 1/80 dilution),

CD44 (Biolegend, cat. No103022, clone IM7, 1/20 dilution), CD62L (Biolegend, cat. No104407, clone MEL-14, 1/100 dilution). All antibodies were diluted according to the instructions and incubated with the cells for 30 min at room temperature. The cells were analyzed by the CytoFLEX S flow cytometry.

**ELISA assay.** H22 tumors were harvested and homogenized on ice in PBS containing protease inhibitor cocktail (Sigma-Aldrich). After centrifugation at $10,000 \times g$ for 10 min at 4 °C twice, the supernatants were collected for the detection of TNF-α, IFN-γ, IL-12p70, and TGF-β activity using ELISA kits according to manufacturer's instruction (DAKEWE, China).

**Collagen I degradation.** Collagen I degradation imaging was performed as previously described[60]. Collagen I was neutralized with 0.01 M NaOH to pH 7.4 and diluted in PBS to 100 μg mL$^{-1}$. Collagen I solution was dropped on the chamber slides, polymerized at 37 °C for 1 h and dried overnight to form compact collagen I films. The collagen I films were biotinylated using 20 μg mL$^{-1}$ EZ-link Sulfo-NHS-LC-LC-biotin in 50 mM NaHCO$_3$ (pH 8.3) for 2 h and then washed with PBS. The collagen I-coated chamber slides were placed in 6-well plates and incubated with PKH26-labeled MPs or Man-MPs at the concentration of 50 μg protein mL$^{-1}$ pretreated with or without 0.6 mg mL$^{-1}$ Bati or 100 μg mL$^{-1}$ AEBSF. After 72 h, the degradation of collagen I was stopped by fixation in 4% paraformaldehyde. The chamber slides were blocked in 2% BSA and stained with streptavidin-FITC (1:200 in 2% BSA). The fluorescence images were visualized by confocal laser scanning microscopy (FV1000, Olympus). Furthermore, 100 μg soluble collagen I was incubated with MPs or Man-MPs at the concentration of 75 μg protein mL$^{-1}$ pretreated with or without 0.6 mg mL$^{-1}$ Bati or 100 μg mL$^{-1}$ AEBSF at 37 °C. After 72-h incubation, the collagen content was measured by Sirius red total collagen detection kit (Chondrex, Redmond, WA, USA).

**SHG imaging.** The tumor tissues were harvested, fixed in 4% paraformaldehyde for 12 h and then dehydrated in 30% sucrose solution. The tumors were frozen in Tissue-Tek OCT compound (Sakura, Torrance, CA) and sectioned into 20 μm slices by a freezing microtome (Leica, Wetzlar, Germany). SHG images were acquired with an LSM 780 NLO upright microscope (Carl Zeiss AG, Oberkochen, Germany) equipped with a W Plan-Apochromat ×20 objective and a mode-locked Ti:Sapphire laser. The tumor slices were observed with polarized laser light at a wavelength of 800 nm and the SHG signals were detected with a 400 nm filter. Images were captured and analyzed using Zen lite 2012 software (Carl Zeiss AG).

**Collagen I staining.** The tumors were harvested, washed with PBS and sectioned by cryotomy. The sections were incubated with Cy3-conjugated collagen I antibody at 37 °C for 30 min, washed with PBS and then the images were acquired by fluorescence microscopy.

**Chemotaxis assay.** Chemotaxis assay was performed using 5-μm transwell filter (Corning Costar). M2-like macrophages were treated with Met@Man-MPs at Met concentration of 20 μg mL$^{-1}$ in the presence or absence of Etan (0.5 μg mL$^{-1}$) for 24 h in the bottom chambers. The lymphocytes isolated from the spleens using Ficoll-Paque PLUS density gradient media were seeded in the top chambers. After 8 h, the suspensions from bottom chambers were collected and the numbers of CD45$^+$CD3$^+$CD8$^+$ T cells were recorded by CytoFLEX S flow cytometry.

**Tumor penetration of anti-PD-1 antibody in vivo.** When tumor volume of H22 tumor-bearing mice reached around 150 mm$^3$, the mice were intravenously injected with PBS, Met@Man-MPs derived from RAW264.7 cells or Met@Man-MPs pretreated with 0.6 mg mL$^{-1}$ Bati at the Met dosage of 10 mg kg$^{-1}$ every other day for two times. Then the mice were intraperitoneally injected with Cy5-labeled anti-PD-1 antibody (100 μg). At 24 h after injection, the mice were sacrificed and tumor tissues were collected, washed with PBS and frozen sectioned. The tumor sections were incubated with FITC-conjugated CD31 antibody (1:200, Biolegend) at 37 °C for 30 min, washed with PBS and then detected using FV1000 confocal microscope. The Cy5 fluorescence was quantified using ImageJ version 2.0.0 software. The penetration capacity of anti-PD-1 antibody was expressed as the ratio of Cy5 in the tumor parenchyma to that within the blood vessels.

**Tumor penetration analysis of anti-PD-1 antibody using the dorsal window chamber tumor model.** Dorsal window chamber imaging was performed as previously described[61]. Briefly, BALB/c mice were anesthetized intraperitoneally with 100 mg kg$^{-1}$ ketamine and 10 mg kg$^{-1}$ xylazine, respectively. Titanium window frames (APJ Trading Co., Inc., Ventura, CA) were installed onto the back of the mice. One side of the skin in the frame was peeled ~1 cm in diameter and the other side remained intact. $5 \times 10^5$ H22 tumor cells were then injected into the fascia of the intact skin. After inoculation of tumor cells, a 10-mm-diameter coverslip was covered and fixed on the peeled skin. When tumors reached ~5 mm in diameter in dorsal window chamber models, the mice were intravenously injected with PBS, Met@Man-MPs derived from RAW264.7 cells, or Met@Man-MPs pretreated with 0.6 mg mL$^{-1}$ Bati at the Met dosage of 10 mg kg$^{-1}$ every 2 days twice. The vasculature was labeled with 50 μL of fluorescein-labeled tomato lectin (Vector

laboratories; 2 mg ml$^{-1}$) 30 min prior to imaging. The tumor images were acquired within 10 min after intravenous injection of 50 µg of Cy5-labeled anti-PD-1 antibody by LSM 780 NLO upright microscope (Carl Zeiss). The fluorescein and Cy5 were excited using 488 and 640 nm laser, respectively. The Cy5 fluorescence intensity was measured by Image J software. The penetration capacity of anti-PD-1 antibody was displayed as the extravascular Cy5 fluorescence normalized by the maximum Cy5 fluorescence within the blood vessels at 10 min post-injection.

**Human organotypic slice culture**. Fresh HCC tissues and blood were obtained from liver cancer patients undergoing routine surgical resection at the Tongji Hospital, Tongji Medical College of Huazhong University of Science and Technology (Wuhan, China). The study was approved by the Clinical Trial Ethics Committee of Huazhong University of Science and Technology and all patients were informed consent. Tumors were cut into thin slices of about $5 \times 5 \times 2$ mm and randomly divided into twelve parts. Human blood was processed to obtain PBMCs by Ficoll gradient. Then the tumor slices were incubated with PBMCs ($1 \times 10^6$ cells mL$^{-1}$) from the same patient in 8 mL RPMI 1640 medium containing 10% FBS in the presence of anti-human PD-1 antibody, Met@Man-MPs derived from THP-1-originated macrophages or Met@Man-MPs plus anti-human PD-1 antibody at the concentration of anti-PD-1 antibody and Met of 20 and 40 µg mL$^{-1}$, respectively. After incubation at 37 °C for 36 h, the tumor slices were washed three times with PBS, minced and digested with 0.8 mg mL$^{-1}$ collagenase type I and 5 µg mL$^{-1}$ DNase I in RPMI 1640 medium at 37 °C for 1 h. The cells were filtered twice using a 40 µm cell strainer and washed twice with PBS. Tumor cells were gated by anti-human CD45 (Biolegend, cat. No304006, clone HI30, 1/20 dilution) negative and stained with anti-Annexin V (Biolegend, cat. No 640920, 1/20 dilution) and PI to analyze the apoptosis. For TAM analysis, cell suspensions were stained with anti-human CD14 (Biolegend, cat. No 301805, clone M5E2, 1/20 dilution), CD11b (Biolegend, cat. No 301309, clone ICRF44, 1/20 dilution), CD80 (Biolegend, cat. No 305217, clone 2D10, 1/20 dilution) and CD206 (Biolegend, cat. No 321125, clone 15-2, 1/20 dilution). For CD8$^+$ T cell infiltration and activation analysis, the cells were stained with anti-human CD45 (Biolegend, cat. No 304006, clone HI30, 1/20 dilution), CD3 (Biolegend, cat. No 344816, clone SK7, 1/20 dilution), CD8a (Biolegend, cat. No 300907, clone HIT8a, 1/20 dilution), CD69 (Biolegend, cat. No 310909, clone FN50, 1/20 dilution), Granzyme B (Biolegend, cat. No 372203, clone QA16A02, 1/20 dilution) or IFN-γ (Biolegend, cat. No 506510, clone B27, 1/20 dilution). All antibodies were diluted according to the instructions and incubated with the cells for 30 min at room temperature. The cells were analyzed by the CytoFLEX S flow cytometry.

**Statistical analysis**. Experiments were repeated at least three times. Statistical analysis was performed using GraphPad Prism 7.0 (GraphPad Software, CA) software. Comparison between two groups was performed using unpaired two-tailed Student's $t$-test. For comparison of multiple groups, one-way ANOVA or two-way ANOVA was used, followed by Tukey's honest significant difference (HSD) post-hoc test or Bonferroni's multiple comparisons post-test, except where otherwise noted. Results are expressed as means ± SD. $P < 0.05$ was considered statistically significant.

**Reporting summary**. Further information on research design is available in the Nature Research Reporting Summary linked to this article.

## Data availability

The authors declare that the main data supporting the findings of this study are available within the article and its Supplementary Information or from the authors upon reasonable request. Source data are provided with this paper.

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

## Acknowledgements

This work was supported by the National Key R&D Program of China (2020YFA0710700 and 2018YFA0208900), National Natural Science Foundation of China (81974459, 81627901, 81773653, and 81672937). We thank the Research Core Facilities for Life Science (HUST), the Analytical and Testing Center of Huazhong University of Science and Technology, Optical Bioimaging Core Facility of WNLO-HUST, and Wuhan institute of biotechnology for related analysis. We thank Shuyan Liang and Zhixin Qiu from Wuhan Biobank Co., Ltd for their kind help in flow cytometric analysis.

## Author contributions

L.G., B.H., and X.Y. designed the project. Z.W., X.Z., T.Y., N.B., G.Z., X.L., Q.L., J.L., J.Y., G.H., Y.Y., and Z.Z. performed the experiments. Z.W., X.Z., B.Z., L.G., B.H., and X.Y. analysed and interpreted the data, and wrote the manuscript.

## Competing interests

The authors declare no competing interests.
