## [Peer Review File · Nature Communications]

Reviewers' comments:

Reviewer #1 (Remarks to the Author):

In the manuscript by Wei, Zhang et al the authors tested the efficacy of mannose- modified macrophage-derived microparticles to repolarize macrophages from M2 to M1 and exert anti tumoral activity.

Mannose-modified microparticles (Man-MPs) uptake was most efficient in macrophages (RAW264.7 and primary-BMDMs) stimulated with IL-4; when compared with untreated and LPS+IFN γ treated counterparts.

In vivo (H22 tumor-bearing mice), Man-MPs accumulated in tumors, when compared to organs (heart, spleen, lung, liver). Within the tumor, they accumulate specifically in 'M2 like macrophages, showing specificity.

Murine IL-4 stimulated macrophages that were treated with Met@Man-MPs, upregulated 'pro-inflammatory macrophage-related genes (Tnfa, iNOs) and proteins (CD80, CD86); and downregulated 'anti-inflammatory macrophage-related genes' (Mrc1, Mgl1, Arg1) and protein (CD206). The authors observed equivalent results on IL-4 treated THP-1 human cells.

Supernatants of Met@Man-MPs treated IL-4 stimulated murine macrophages transferred to HepG2 and murine 4T1 tumor cells showed the most decreased tumor cell-viability when compared with other controls (Secreted factors of TAMs promote tumor cell viability).

Met@Man-MPs treatment inhibited tumor growth in vivo (volume and weight), and delayed tumor formation. No body weight change, effects on major organs, or serological changes were observed, so Met@Man-MPs caused no obvious toxicity. As a mechanism, Met@Man-MPs caused an increase in the number of tumor 'M1-like macrophages', CD8+, CD8+CD69+, CD4+ and CD4+CD69+Tcells in tumors, while it caused a decrease in 'M2-like macrophages, MDSCs and T-regs. There was an increase in inflammatory cytokines TNF- α , IFN- γ and IL12p70; while there was a decrease in TGF β . By using liposomal clodronate, they claimed that the observed effects were dependent on macrophages.

Man-MPs degrade tumor collagen: both murine and mouse Man-MPs have high MMP activity and could degrade collagen in a concentration dependent manner (in vitro). In vivo, Man-MPs decreased the deposition of collagen fibre network. Met@Man-MPs treated M2 like macrophages can recruit CD8+Tcells in vitro in a TNF- α dependent manner. Degrading of collagen also contributes to infiltration, because treatment of mice with Met@Man-MPs and Bati (an MMP activity inhibitor), as well as treatment with Met@Man-MPs and Etan (TNF- α inhibitor) abrogated the infiltration. So, TNF- α and degradation of collagen are both involved in infiltration.

Met@Man-MPs enhance accumulation and penetration of PD-1 antibody in tumors, and again, treatment with Bati abrogated the effect (collagen is involved in this effect). Met@Man-MPs improve the anticancer activity of PD-1 antibody, as treatment with both have a significantly higher anticancer activity with no significant toxicity. Combined treatment led to increased survival, when compared to monotherapies. In a tumor-re-challenge model, combined treatment induced long-term antitumor effect (immunological memory). To validate this in humans, they used a slice culture model (liver-cancer patient tumor derived) and PBMCs from the same patients.

Combination treatment with Met@Man-MPs and anti-PD1 led to strongest tumor cell apoptosis, and CD8+T cell activation.

In summary, Met@Man-MPs causes re-polarization of TAMs and increases the recruitment of CD8+ T-cells via macrophage secreted factors and collagen degradation mechanisms. At the same time, it decreases the accumulation of immune suppressive cells (MDSCs and Tregs). Due to the Man-MPs ability to degrade collagen, therapeutic antibodies, like anti-PD-1 can infiltrate the tumor better. Combination treatment of anti-PD-1 and Met@Man-MPs led to increased anti-cancer activity, long-term memory immunity and survival of mice.

Major comments:

A. Can the particles be prepared using primary macrophages (both mouse and human) instead of

cell lines? Do they have the same efficacy?

B. Where do the particles go when internalized? In the lysosomes? How are they internalized? How is metformin taken up by macrophages? Is the M2 to M1 switch mediated by the same mechanism reported by the 2 previous studies cited in the manuscript

(<https://www.ncbi.nlm.nih.gov/pmc/articles/PMC4742188/pdf/oncotarget-06-36441.pdf>

<https://www.ncbi.nlm.nih.gov/pmc/articles/PMC5400538/pdf/oncotarget-08-20706.pdf>)

C. All the paper is based on the assumption that there is a net dichotomy between M1 and M2 in the tumor microenvironment, being TAM M2-like. However different studies profiling the human tumor microenvironment are now reporting that such dichotomy is not so well defined, as TAM express a spectrum of M1 and M2 markers. What are the gating strategies to designate the cells in the tumor as M1 TAMs, M2 TAMs, etc? This is a problem of using the dichotomy of M1 and M2 and combining this nomenclature in both in vitro and in vivo. Do M2 TAMs express the mannose receptor before treatment? There is no data showing that. This data would also strengthen the data on 2g, if they show that 'M2' macrophages in tumors have the highest expression of the mannose receptor when compared to all other macrophage subsets. Moreover in Fig 2F there is a distinction of M1 and M2 macrophages, while in Fig 2G this population becomes "TAM" without the indication of the markers used to define it.

D. Did you perform a time scale experiment to determine which is the limit of detection of the particles? Is 4h post injection the peak of detection in the tumor?

E. Were the particles used in the in vivo experiments only derived from the macrophage cell line RAW264.7?

F. Figure S7B: How was viability measured? Why are there viability values over 100%?

G. The authors show that the supernatant of Met@Man-MPs lead to a reduced viability of tumor cells.

a. Could they do a supernatant profile to see the differentially secreted factors and generate candidates that mediate the loss of viability in tumor cells?

b. What about the cell-cell contact effect? Do Met@Man-MPs exert further tumoricidal function i.e Phagocytosis, ADCC?

H. Fig 3 is a bit confusing as I would expect Met to behave similarly to the Met@MPs in terms of polarization, but it does not seem the case. Moreover the authors do not show any mechanism on how this switch is caused by Met@MPs compared to free Met.

I. The in vivo studies are performed using a subcutaneous tumor model (H22) or an orthotopic breast cancer model (4T1); both models are very artificial and not spontaneous; I would like to see the key experiments repeated in a relevant spontaneous tumor model to strengthen the results.

J. Is the polarization switch occurring on primary human macrophages as well? Can the authors show the same experiment shown in Fig 3 using primary MDMs and Met@MPs derived from MDMs?

K. Ex vivo results from human cancer (fig9): you need to show the gating strategies used to identify the macrophages. Are those the only 2 populations identified? How can you be sure you are measuring only tumor apoptosis and not other cells apoptosis as well? What is the effect of PBMC in this experiment. I would like to see the same experiment without PBMC incubation, to prove that only when PBMC are added, the changes occur (or not).

L. The discussion is too short. What about other studies, has anyone used these MPs in this kind of setting? What are the implications and future directions? How do the authors envisage the improvement of the technology?

Minor comments:

1. Line 112. There should be a statement like M2 macrophages highly express the mannose receptor and then point to Supplementary Figure 1. The text as it is, doesn't really correlate with that is shown in the figure.

2. Supplementary figure 1, why is it %? Shouldn't it be Fold over? What are the data points? It is not clear

3. Mrc1 is misspelled as Mcr1, line 176.

4. When there is the claim that Met@Man-MPs repolarize macrophage, it is not clear when they did gene expression vs. protein expression experiments. It would be better if these were clarified, especially when they treat Mrc1 and CD206 as independent, it is not clear that Mrc1 encodes CD206 (Mannose receptor)

Reviewer #2 (Remarks to the Author):

The manuscript by Drs Wei et al. entitled "Boosting Anti-PD-1 Therapy by Metformin-Loaded Targeting Microparticles Derived from Macrophages" deals with development of mannose-modified macrophage-derived microparticles (Man-MPs) loaded metformin (Met@Man-MPs) target to M2-like TAMs and polarize them towards M1-like phenotype for improved immunotherapy. This is a comprehensive and well written study on an important field. However, it can be improved by addressing the concerns below.

Major points:

1. To conclusively demonstrate cellular targeting profile of Man-MPs in tumors, authors need to provide confocal images of tissue sections with labeled particles in the context of cell type specific markers. Fig. 2f data on cellular distribution were obtained from minced tumor incubated for 30 min at 37C (L566), this may have led to unphysiological shifts in cellular distribution. In addition, biodistribution of MPs in tissue sections of organs known to express mannose receptors (e.g. liver) should be studied by confocal microscopy in the context of mannose receptors CD206 and CD209.
2. Tumor implantation site has profound effect on tumor status, including on vascular function and leakiness (for example, with orthotopic models typically showing more normal and less leaky blood vessels). Authors should provide rationale for their using the s.c. H22 model and perform at least a minimum set of experiments to show the validity of their observations in orthotopic model.
3. Marimastat is a first generation of MMP inhibitor based on the structure of the collagen molecule. It should be mentioned that hydroxamate activity-based MMP probes related to marimastat bind many off-target metalloproteinases that are not MMPs (Proc Natl Acad Sci USA. 2004;101:10000-10005).
4. The figures are very busy and would benefit from moving nonessential graphs into the SI section.

Other points.

Introduction

- L95-96 statement "macrophage derived MPs have natural tumor targeting capacity" is not supported by references used to support the statement. Ref 37 is a review paper that does not mention microparticles and Ref 38 deals with development of exosome-like macrophage-derived nanovesicles.
- Discussion on translational potential of microparticles (including large scale manufacturing feasibility) and ongoing clinical trials needs to be included.
- The authors should avoid talking generally about "mannose receptor" (e.g. in L96), L137) and always mention the receptor type. Mannose and its analogues are not specific for CD206: they also bind other mannose receptors, such as CD209 expressed in the skin and intestinal and genital

mucosa.

Figures and Results.

Fig. 1.

- A – Include labeling of antibody: “Anti PD1”
- D - Include lower mag views to allow appreciating the larger population of MPs

Fig. 3.

- F-H - Start Y axis at 40% viability for better visualization of the effect. Why is “cells” added for H22 and 4T1 and not for HepG2?

Fig. 5..

- A – should be moved to SI. It is unclear why confocal is used in this cell free assay. Collagen layer appears non-uniform making one to question the assay and think of alternatives (such as double quenched collagen assays). Epifluorescence microscopy and using higher magnification to better see the MPs would give better understanding of fate of collagen layer upon incubation with the MPs. Labeling should be used to point to areas of degradation. High dose MP samples should be treated also with optional inhibitors of MMPs and serine proteases (as it is likely that serine proteases such as plasminogen activation system is involved in the activation of collagen degrading proteases).
- B – Early time point should be moved to SI, it adds clutter with minimal additional information. High-dose MP samples should be treated also with optional inhibitors of MMPs and serine proteases (as it is likely that serine proteases such as plasminogen activation system is involved in the activation of collagen degrading proteases).
- C,D – Generally, different tumor areas show greatly varying deposition of ECM and for conclusive data it is essential to include a low magnification view of the entire tumor. It is unclear if n=10 refers to independent tumor mice or fields analysed, please specify.
- E,F – Same comment and for CD on magnification – to be convincing, larger areas need to be shown. Authors should comment on collagen1 staining pattern and lack of fibrillary structures.

Fig. 6

- B – indicate the tumor margin by a dotted line and add additional tissue context by performing CD31 staining (to allow relating CD8+ cell and vascular densities. A low mag view of entire tumor with CD8+ cells highlighted would be more convincing).
- C,D – authors should comment on surprisingly smaller effect of Met@ManMPs on inducing CD8+ cell number in tumors in histology based analysis of tumor interior (C, left) compared to total tumor (D). It would be expected that the difference is larger at the center, as it is not diluted by margin with no difference.

Fig. 7

Here and elsewhere would be reader-friendly to use color-coded text to explain labeling.

- C - Accumulation of anti PD1 antibody should be studied in the context of vascular permeability. At minimum, the tumors should be stained for endogenous mouse IgG as a surrogate marker as vascular permeability.
- D – number of mice used to generate the data points needs to be specified (information on number of fields is provided).
- E – Include labeling (arrows) to point to extravascular PD1 signal. It is not mentioned that lectin was used as a vascular tracer, add.
- F - number of mice used to generate the data points needs to be specified.

Materials and methods

- L619-620: please specify how were the metastatic nodules counted? On tissue sections based on H&E stain, macroscopically??
- L723: Is it correct that tumor slices for organotypic culture were 5 mm (0,5 cm) thick? This is

very thick and poses a serious diffusion problem for the gas and nutrient exchange. Authors should correct and/or comment on that.

RESPONSES TO REVIEWERS

We would like to express our sincere thanks to all two reviewers for their critical and constructive comments. We have performed substantial additional experiments to address their concerns. We respond point-by-point to each of their comments and criticisms. We feel that their comments have helped us on significantly improving and strengthening the manuscript and clarifying some issues. We hope that the revision has addressed their major concerns.

Response to Reviewer 1

In the manuscript by Wei, Zhang et al the authors tested the efficacy of mannose-modified macrophage-derived microparticles to repolarize macrophages from M2 to M1 and exert antitumoral activity.

Mannose-modified microparticles (Man-MPs) uptake was most efficient in macrophages (RAW264.7 and primary-BMDMs) stimulated with IL-4; when compared with untreated and LPS+IFN γ treated counterparts.

In vivo (H22 tumor-bearing mice), Man-MPs accumulated in tumors, when compared to organs (heart, spleen, lung, liver). Within the tumor, they accumulate specifically in 'M2 like macrophages, showing specificity.

Murine IL-4 stimulated macrophages that were treated with Met@Man-MPs, upregulated 'pro-inflammatory macrophage-related genes (Tnfa, iNOs) and proteins (CD80, CD86); and downregulated 'anti-inflammatory macrophage-related genes' (Mrc1, Mgl1, Arg1) and protein (CD206). The authors observed equivalent results on IL-4 treated THP-1 human cells. Supernatants of Met@Man-MPs treated IL-4 stimulated murine macrophages transferred to HepG2 and murine 4T1 tumor cells showed the most decreased tumor cell-viability when compared with other controls (Secreted factors of TAMs promote tumor cell viability).

Met@Man-MPs treatment inhibited tumor growth in vivo (volume and weight), and delayed tumor formation. No body weight change, effects on major organs, or serological changes were observed, so Met@Man-MPs caused no obvious toxicity. As a mechanism, Met@Man-MPs caused an increase in the number of tumor 'M1-like macrophages', CD8⁺, CD8⁺CD69⁺, CD4⁺ and CD4⁺CD69⁺ T cells in tumors, while it caused a decrease in 'M2-like macrophages, MDSCs and T-regs. There was an increase in inflammatory cytokines TNF- α , IFN- γ and IL12p70; while there was a decrease in TGF β . By using liposomal clodronate, they claimed that the observed effects were dependent on macrophages.

Man-MPs degrade tumor collagen: both murine and mouse Man-MPs have high MMP activity and could degrade collagen in a concentration dependent manner (in vitro). In vivo, Man-MPs decreased the deposition of collagen fibre network. Met@Man-MPs treated M2 like macrophages can recruit CD8⁺ T cells in vitro in a TNF- α dependent manner. Degrading of collagen also contributes to infiltration,

because treatment of mice with Met@Man-MPs and Bati (an MMP activity inhibitor), as well as treatment with Met@Man-MPs and Etan (TNF-alpha inhibitor) abrogated the infiltration. So, TNF-alpha and degradation of collagen are both involved in infiltration.

Met@Man-MPs enhance accumulation and penetration of PD-1 antibody in tumors, and again, treatment with Bati abrogated the effect (collagen is involved in this effect). Met@Man-MPs improve the anticancer activity of PD-1 antibody, as treatment with both have a significantly higher anticancer activity with no significant toxicity. Combined treatment led to increased survival, when compared to monotherapies. In a tumor-re-challenge model, combined treatment induced long-term antitumor effect (immunological memory). To validate this in humans, they used a slice culture model (liver-cancer patient tumor derived) and PBMCs from the same patients. Combination treatment with Met@Man-MPs and anti-PD1 led to strongest tumor cell apoptosis, and CD8+T cell activation.

In summary, Met@Man-MPs causes re-polarization of TAMs and increases the recruitment of CD8+ T-cells via macrophage secreted factors and collagen degradation mechanisms. At the same time, it decreases the accumulation of immune suppressive cells (MDSCs and Tregs). Due to the Man-MPs ability to degrade collagen, therapeutic antibodies, like anti-PD-1 can infiltrate the tumor better. Combination treatment of anti-PD-1 and Met@Man-MPs led to increased anti-cancer activity, long-term memory immunity and survival of mice.

Major comments:

(1) Can the particles be prepared using primary macrophages (both mouse and human) instead of cell lines? Do they have the same efficacy?

Response: We thank the reviewer's constructive suggestion. In the revised manuscript, we prepared mannose-modified MPs derived from mouse bone marrow-derived macrophages (BMDMs) and human peripheral blood monocyte-derived macrophages (MDMs) to load metformin (Met). Our results showed that consistent with the Met@Man-MPs derived from RAW264.7 cells and THP-1-derived macrophages, Met@Man-MPs derived from BMDMs showed the strongest capacity to enhance the expression of M1-related markers and decrease the expression of M2-related markers in IL-4-conditioned BMDMs, even better than high dosage of free Met. Consistently, Met@Man-MPs derived from human MDMs exhibited the strongest M2-like macrophage repolarization activity in IL-4-conditioned human MDMs. These results suggested that M2-like macrophage repolarization activity of Met@Man-MPs derived from macrophages of different origins is universal. We added the supplemented data in the revised manuscript, Supplementary Figures 13 and 15.

(2) Where do the particles go when internalized? In the lysosomes? How are they internalized? How is metformin taken up by macrophages? Is the M2 to M1 switch mediated by the same mechanism reported by the 2 previous studies cited in the manuscript

Response: We thank the reviewer's constructive question. In the revised manuscript, we determined the endocytic pathway of MPs and Man-MPs using different endocytic inhibitors. Our results demonstrated that caveolin-, clathrin-dependent endocytosis and phagocytosis were involved in the cellular uptake of MPs and Man-MPs in M2-like macrophages. To further determine the intracellular trafficking of Met@MPs or Met@Man-MPs in M2-like macrophages, we used Rhodamine B (RhB) as a model drug to load into DiO-labelled MPs and Man-MPs. Confocal microscopic images clearly showed that MPs/Man-MPs and RhB entered M2-like macrophages together and then colocalized with lysosomes after internalization, followed by drug release over time. We added the supplemented data in the revised manuscript, Supplementary Figures 17 and 18.

Met has been reported to repolarize M2-like TAMs to M1 phenotype through AMPK signaling pathway. To determine whether Met@Man-MPs used the same mechanism to repolarize M2-like TAMs, M2-like macrophages were treated with Met@Man-MPs or high dosage of Met (as a control) in the presence or absence of compound C (CC, an AMPK inhibitor), and then the mRNA levels of M1- and M2-related markers were determined by RT-PCR. Our results showed that Met@Man-MPs significantly increased the expression of M1-related markers (*TNF- α* and *iNOS*) and decrease the expression of M2-related markers (*Mrc1* and *Arg1*), even better high dosage of Met. However, similar to high dosage of Met, Met@Man-MPs-induced increase in M1-related markers and decrease in M2-related markers were significantly abrogated by CC treatment. These results suggest that Met@Man-MPs, like free Met, repolarized M2-like macrophages to M1 phenotype through AMPK signaling pathway. We added the supplemented data in the revised manuscript, Supplementary Figure 16.

(3) All the paper is based on the assumption that there is a net dichotomy between M1 and M2 in the tumor microenvironment, being TAM M2-like. However different studies profiling the human tumor microenvironment are now reporting that such dichotomy is not so well defined, as TAM express a spectrum of M1 and M2 markers. What are the gating strategies to designate the cells in the tumor as M1 TAMs, M2 TAMs, etc? This is a problem of using the dichotomy of M1 and M2 and combining this nomenclature in both in vitro and in vivo. Do M2 TAMs express the mannose receptor before treatment? There is no data showing that. This data would also strengthen the data on 2g, if they show that 'M2' macrophages in tumors have the highest expression of the mannose receptor when compared to all other macrophage subsets. Moreover in Fig 2F there is a distinction of M1 and M2 macrophages, while in Fig 2G this population becomes "TAM" without the indication of the markers used to define it.

Response: We thank the reviewer's constructive suggestion. We agreed with the reviewer that the actual polarization state of macrophages is far more complex than the simple dichotomy M1/M2 classification. However, it is largely accepted that most TAMs lie on a spectrum that can encompass features of both classically activated 'M1' macrophages exhibiting tumor-inhibitory and immune-stimulatory functions, and

alternatively activated ‘M2’ macrophages exhibiting tumor-promoting and immune-suppressive functions (*Nature Reviews Immunology*, 2019;19:369-382; *Nat Rev Clin Oncol*, 2017, 14:399-416). In our original manuscript, we gated M1-like TAMs as CD11b⁺F4/80⁺CD11c⁺CD206⁻ cells and M2-like TAMs as CD11b⁺F4/80⁺CD206⁺CD11c⁻ cells according to the reported literatures (*Proc Natl Acad Sci U S A*, 2016, 113:4470-4475; *Nat Cell Biol* 2016, 18:790-802; *Nature*, 2016, 539:443-447). Considering that the markers for M1 and M2 phenotypes can have partial overlap due to the incomplete polarization of individual macrophages (*Sci Transl Med*, 2019, 11: eaat1500), we changed the previous gating and used CD11b⁺F4/80⁺CD11c⁺ cells for M1-like TAMs and CD11b⁺F4/80⁺CD206⁺ cells for M2-like TAMs as (*Nature*, 2019, 572: 392-396; *Nat Commun*, 2019, 10:2272; *Nat Commun*, 2020, 11:4064). This altered gating strategy did not affect the conclusion of M2-like TAM targeting of Man-MPs and the repolarization of M2-like TAMs to M1 phenotype by Met@Man-MPs. The gating strategy was shown in Supplementary Figure 6 in the revised manuscript.

In the original manuscript, we found that the mRNA expression of *Mrc1* (encoding mannose receptor CD206/MRC1) in M2-like macrophages (IL-4-conditioned RAW264.7 cells) was 255.5 fold of M0 macrophages (RAW264.7 cells) and 149.5 fold of M1-like macrophages (LPS- and IFN- γ -conditioned RAW264.7 cells), suggesting the high mannose receptor expression in M2-like macrophages. Man-MPs were mainly accumulated in M2-like TAMs (CD11b⁺F4/80⁺CD206⁺ cells) compared with GFP⁺ tumor cells and other immune cells, including M1-like TAMs (CD11b⁺F4/80⁺CD11c⁺ cells), DCs (CD45⁺F4/80⁻CD11c⁺ cells), CD4⁺ T cells (CD45⁺CD3⁺CD4⁺ cells), CD8⁺ T cells (CD45⁺CD3⁺CD8a⁺ cells), MDSCs (CD45⁺CD11b⁺Gr1⁺ cells) and Tregs (CD45⁺CD4⁺CD25⁺FoxP3⁺ cells) in tumor tissues (Figure 2f in the revised manuscript), suggesting M2-like TAM targeting of Man-MPs. Furthermore, more Man-MPs were found to be accumulated in the total TAMs compared with liver Kupffer cells, splenic macrophages, pulmonary macrophages (all these macrophages including TAMs were gated as CD11b⁺F4/80⁺ cells) and splenic DCs (CD45⁺F4/80⁻CD11c⁺ cells) (Figure 2g in the revised manuscript), excluding the targeting accumulation of Man-MPs in other tissue-resident macrophages and phagocytes. To clarify the possible reason, in the revised manuscript, we determined CD206 expression in total TAMs and other tissue-resident macrophages and phagocytes. Our results showed that CD206 expression was higher in TAMs than other cells. In view of the tumor-targeting capacity of macrophages-derived MPs and high CD206 expression in TAMs, it was reasonable that more Man-MPs were accumulated in TAMs, especially M2-like TAMs (compared with M1-like TAMs and other immune cells as seen in Figure 2f in the revised manuscript). We added the supplemented data in the revised manuscript, Supplementary Figure 8.

(4) Did you perform a time scale experiment to determine which is the limit of detection of the particles? Is 4h post injection the peak of detection in the tumor?

Response: In this study, we determined the biodistribution of MPs or Man-MPs in H22 tumor-bearing mice at different time intervals after intravenous injection of Cy5-labelled MPs or Man-MPs (**Figure 1**). Our results showed that Man-MPs exhibited stronger tumor accumulation at different time points. The highest tumor accumulation of MPs and Man-MPs was shown at 24 h after injection (We just put the data at 24 h after injection in the manuscript). In view of this fact, we determined the accumulation of Man-MPs in tumor and immune cells of tumor tissues (Figure 2f in the revised manuscript) at 24 h after injection. All these data showed that Man-MPs exhibited good M2-like TAM-targeting capability.

Figure 1. Ex vivo imaging of Cy5 in the organs and tumors of H22 tumor-bearing mice at different time intervals after intravenous injection of Cy5-labelled MPs or Man-MPs at the dosage of 15 mg protein kg^{-1} . Data are presented as means \pm s.d. (n=4). * $P < 0.05$, *** $P < 0.001$.

(5) Were the particles used in the in vivo experiments only derived from the macrophage cell line RAW264.7?

Response: We thank the reviewer's insightful question. In the in vivo experiments, we used murine H22 subcutaneous tumor model, 4T1 orthotopic breast tumor model and AOM/DSS-induced colitis-associated cancer (CAC) mouse model to determine the anticancer activity of Met@Man-MPs with or without anti-PD-1 antibody and the improved tumor immune microenvironment. To avoid the immune response, we used Met@Man-MPs derived from murine RAW264.7 macrophages to treat these tumor-bearing mice. However, in the experiments of evaluating the anticancer and immune microenvironment using organotypic ex vivo slices from liver cancer patient-derived tumors, we used Met@Man-MPs derived from human monocytic THP-1-originated macrophages to treat these tumor slices. All these data showed that Met@Man-MPs exhibited good anticancer and improve tumor immune microenvironment to synergize the effects of anti-PD-1 antibody.

(6) Figure S7B: How was viability measured? Why are there viability values over 100%?

Response: In the original Figure S7 (Figure S11 in the revised manuscript), we tried to confirm that IL-4-conditioned RAW264.7 exhibited the characteristics of M2-like macrophages. Since M2-like macrophages have the ability to promote tumor growth, we determined the cell viability of H22, 4T1 and HepG2 cells after treatment with the supernatants of RAW264.7 cells or IL-4-conditioned RAW264.7 for 24 h by CCK-8 assay. Our results confirmed that the supernatants of IL-4-conditioned RAW264.7 cells significantly increased the cell viability of these tumor cells. Here, we set the cell viability of tumor cells treated with the supernatants of RAW264.7 cells as 100%. Thus, the cell viability of tumor cells treated with the supernatants of IL-4-conditioned RAW264.7 cells exceeded over 100%.

(7) The authors show that the supernatant of Met@Man-MPs lead to a reduced viability of tumor cells.

a. Could they do a supernatant profile to see the differentially secreted factors and generate candidates that mediate the loss of viability in tumor cells?

Response: We thank the review's insightful question. M1-like macrophages exhibit anticancer activity by producing pro-inflammatory cytokines, such as TNF- α and IFN- γ , et al. In the work, we found that Met@Man-MPs efficiently increased TNF- α expression in M2-like macrophages (Figure 3a, Figures S12a, 13a, 14a and 15a in the revised manuscript). Furthermore, the highest TNF- α contents were detected in the supernatants of Met@Man-MPs-treated M2-like macrophages compared with those of MPs-, Man-MPs-, free Met-, Met@MPs-, Met@Man-MPs-, or high dosage of Met-treated group by ELISA assay. To confirm whether the secreted TNF- α in the supernatants of Met@Man-MPs-treated M2-like macrophages was responsible for the reduced viability of tumor cells, we determined the cell viability of H22 and 4T1 cells after incubation with the supernatants of M2-like macrophages treated with Met@Man-MPs in the presence or absence of etanercept (Etan), an inhibitor of TNF- α . Consistently, the supernatants of Met@Man-MPs-treated M2-like macrophages significantly decreased the cell viability of H22 and 4T1 cells. However, Etan markedly abrogated the Met@Man-MPs-induced decrease in cell viability, suggesting that the secreted TNF- α in Met@Man-MPs-treated M2-like macrophages was responsible for the decrease in cell viability of tumor cells. We added the supplemented data in the revised manuscript, Supplementary Figure 19.

b. What about the cell-cell contact effect? Do Met@Man-MPs exert further tumoricidal function i.e Phagocytosis, ADCC?

Response: We thank the reviewer's constructive suggestion. In the revised manuscript, to determine whether Met@Man-MPs-treated M2-like macrophages exhibited tumoricidal function by phagocytosis of tumor cells, CFSE-labeled M2-like macrophages treated with PBS, MPs, Man-MPs, free Met, Met@MPs, Met@Man-MPs, or high dosage of free Met were co-cultured with DiD-labeled H22 or 4T1 cells at a ratio of 1:1 at 37 °C for 3 h, and then the phagocytosis of tumor cells by macrophages was detected as the ratio of the numbers of CFSE⁺DiD⁺ cells to CFSE⁺ cells by flow cytometry. The results showed that Met@Man-MPs-treated

M2-like macrophages possessed the strongest capacity of phagocytosis of tumor cells, suggesting that Met@Man-MPs might exhibit anticancer activity by repolarization of M2-like macrophages to M1 phenotype to phagocytose tumor cells. We added the supplemented data in the revised manuscript, Supplementary Figure S20.

(8) Fig 3 is a bit confusing as I would expect Met to behave similarly to the Met@MPs in terms of polarization, but it does not seem the case. Moreover the authors do not show any mechanism on how this switch is caused by Met@MPs compared to free Met.

Response: In the original manuscript (Figure S4 in the revised manuscript), we detected Met content in M2-like macrophages treated with free Met, Met@MPs or Met@Man-MPs. Our results showed that more Met@MPs were internalized into M2-like macrophages compared with free Met, and the strongest cellular uptake was detected in Met@Man-MPs-treated group. In view of this fact, Met@MPs and Met@Man-MPs exhibited stronger M2-like macrophage repolarization activity than free Met.

(9) The in vivo studies are performed using a subcutaneous tumor model (H22) or an orthotopic breast cancer model (4T1); both models are very artificial and not spontaneous; I would like to see the key experiments repeated in a relevant spontaneous tumor model to strengthen the results.

Response: We thank the reviewer's constructive suggestion. In the revised manuscript, we constructed azoxymethane (AOM)/dextran sodium sulfate (DSS)-induced colitis-associated cancer (CAC) mouse model to evaluate the anticancer activity and tumor immune microenvironment induced by combination of Met@Man-MPs and anti-PD-1 antibody. Our results showed that Met@Man-MPs exhibited good anticancer activity by counting the tumor numbers in colons and H&E staining in colon tumors. Combination of Met@Man-MPs and anti-PD-1 antibody generated the strongest synergistic tumor inhibition. Meanwhile, tumor immune microenvironment analysis showed that Met@Man-MPs significantly increased the percentages of M1-like TAMs, CD8⁺ T cells, activated CD8⁺ T cells and activated CD4⁺ T cells, while decreased the percentages of M2-like TAMs, MDSCs and Tregs. However, combination of Met@Man-MPs and anti-PD-1 antibody exhibited the strongest capacity to reshape tumor immune microenvironment. These data in CAC mice were consistent with those in subcutaneous H22 tumor-bearing mice and orthotopic 4T1 tumor-bearing mice. We added the supplemented data in the revised manuscript, Figure 9 and Supplementary Figures 38-40.

(10) Is the polarization switch occurring on primary human macrophages as well? Can the authors show the same experiment shown in Fig 3 using primary MDMs and Met@MPs derived from MDMs?

Response: We thank the reviewer's constructive suggestion. In the revised manuscript, we prepared mannose-modified MPs derived from human peripheral

blood monocyte-derived macrophages (MDMs) to load metformin (Met). Our results showed that consistent with the data in IL-4-conditioned RAW264.7, BMDMs and THP-1-derived macrophages, Met@Man-MPs derived from human MDMs showed the strongest capacity to repolarize M2-like macrophages to M1 phenotype in IL-4-conditioned MDMs, as evidenced by the enhanced expression of M1-related markers (*TNF α* , *iNOS*, *CD80* and *CD86*) and decreased expression of M2-related markers (*Mrc1* and *CD163*). We added the supplemented data in the revised manuscript, Supplementary Figure 15.

(11) Ex vivo results from human cancer (fig9): you need to show the gating strategies used to identify the macrophages. Are those the only 2 populations identified? How can you be sure you are measuring only tumor apoptosis and not other cells apoptosis as well? What is the effect of PBMC in this experiment. I would like to see the same experiment without PBMC incubation, to prove that only when PBMC are added, the changes occur (or not).

Response: We thank the reviewer's insightful question. In the revised manuscript, we provided the gating strategies used to identify the macrophages in human tumor slices, as shown in Supplementary Figure 41. The M1- and M2-like macrophages were gated as CD11b⁺CD14⁺CD80⁺ cells and CD11b⁺CD14⁺CD206⁺, respectively.

In this manuscript, we used organotypic slices from two liver cancer patient-derived tumors to determine the anticancer activity and improved tumor immune microenvironment induced by combination of Met@Man-MPs and anti-PD-1 antibody. The two independent experiments showed that Met@Man-MPs significantly induced apoptosis in tumor cells, and reshaped the tumor immune microenvironment, including the increased M1-like TAM percentages, the decreased M2-like TAM percentages, and the enhanced infiltration and activation of CD8⁺ T cells in tumor tissues. Combination of Met@Man-MPs and anti-PD-1 antibody resulted in the stronger apoptosis induction in tumor cells and CD8⁺ T cell activation.

We determined the apoptosis induction in tumor cells by gating CD45⁻ cells according to some reported literatures (*Sci Transl Med*, 2019, 11(474):eaat5690; *Nat Biomed Eng*, 2020, 4(7):743-753). The detailed information about determining apoptosis induction in tumor cells was shown in the Methods section.

After evaluating the anticancer activity and tumor immune microenvironment induced by combination of Met@Man-MPs and anti-PD-1 antibody in tumor-bearing mouse models, we further used organotypic slices from liver cancer patient-derived tumors to confirm these effects. Due to the few lymphocytes in these clinical tumor tissues, we did not detect CD8⁺ T cells in the pre-experiments. To clarify the role of enhanced recruitment and tumor infiltration of CD8⁺ T by Met@Man-MPs-educated M2-like TAMs in enhancing the anticancer activity of anti-PD-1 antibody, we added PBMCs in the tumor slices and then determined the apoptosis induction in tumor cells and the activation of CD8⁺ T cells. Similar treatment was reported in some literatures (*Nat Commun*, 2019, 10: 2416; *Cell*, 2018, 174(6):1586-1598).

(12) The discussion is too short. What about other studies, has anyone used these MPs in this kind of setting? What are the implications and future directions? How do the authors envisage the improvement of the technology?

Response: We thank the review's constructive suggestion. In the revised manuscript, we added some discussion about the possible improvement of the technology and translational potential in the Discussion section.

Minor comments:

(13) Line 112. There should be a statement like M2 macrophages highly express the mannose receptor and then point to Supplementary Figure 1. The text as it is, doesn't really correlate with that is shown in the figure.

Response: We thank the review's constructive suggestion. We have revised these sentences in the revised manuscript.

(14) Supplementary figure 1, why is it %? Shouldn't it be Fold over? What are the data points? It is not clear.

Response: We thank the review for pointing out the mistake we made. In fact we determined that the mRNA expression of *Mrc1* in M2-like macrophages (IL-4-conditioned RAW264.7 cells) was 255.5 fold of M0 macrophages (RAW264.7 cells) by RT-PCR. Therefore, the ordinate unit should be fold, not %. We revised the mistake in the revised manuscript. The data points were from the results of four independent experiments.

(15) *Mrc1* is misspelled as *Mcr1*, line 176.

Response: We thank the review for pointing out the mistake we made. We changed the misspelled *Mrc1* to *Mcr1* in the revised manuscript.

(16) When there is the claim that Met@Man-MPs repolarize macrophage, it is not clear when they did gene expression vs. protein expression experiments. It would be better if these were clarified, especially when they treat *Mrc1* and CD206 as independent, it is not clear that *Mrc1* encodes CD206 (Mannose receptor).

Response: We thank the review's constructive suggestion. In the revised manuscript, we emphasized the changes of gene and protein expression of M1- and M2-related markers in evaluating the repolarization of M2-like macrophages by Met@Man-MPs. Meanwhile, we emphasized that *Mrc1* gene encodes mannose receptor CD206/MRC1.

Response to Reviewer 2

The manuscript by Drs Wei et al. entitled “Boosting Anti-PD-1 Therapy by Metformin-Loaded Targeting Microparticles Derived from Macrophages” deals with development of mannose-modified macrophage-derived microparticles (Man-MPs) loaded metformin (Met@Man-MPs) target to M2-like TAMs and polarize them towards M1-like phenotype for improved immunotherapy. This is a comprehensive and well written study on an important field. However, it can be improved by addressing the concerns below.

Major points:

1. To conclusively demonstrate cellular targeting profile of Man-MPs in tumors, authors need to provide confocal images of tissue sections with labeled particles in the context of cell type specific markers. Fig. 2f data on cellular distribution were obtained from minced tumor incubated for 30 min at 37°C (L566), this may have led to unphysiological shifts in cellular distribution. In addition, biodistribution of MPs in tissue sections of organs known to express mannose receptors (e.g. liver) should be studied by confocal microscopy in the context of mannose receptors CD206 and CD209.

Response: We thank the reviewer’s constructive suggestion. In the revised manuscript, we supplemented the confocal microscopic images of tumors in H22 tumor-bearing mice at 24 h after intravenous injection of PKH26-labelled MPs or Man-MPs (red) and then labelled with fluorescence-conjugated antibodies to CD86 (labelling M1-like TAMs), CD206 (labelling M2-like TAMs), CD4 (labelling CD4⁺ T cells), CD8 (labelling CD8⁺ T cells), Gr1 (labelling MDSCs) and FoxP3 (labelling Tregs). The overlay of PKH26 and immune cells showed that more Man-MPs were colocalized with M2-like TAMs compared with other immune cells in tumor tissues, confirming the good M2-like TAM-targeting capability of Man-MPs. We added the supplemented the data in the revised manuscript, Supplementary Figure 7.

In Figure 2f, we tried to determine the M2-like TAM targeting of Man-MPs. To quantitatively determine the accumulation of Man-MPs in tumor and immune cells, the tumor tissues were cut into small pieces and then incubated in RPMI 1640 medium containing 0.8 mg mL⁻¹ collagenase type I and 5 μg mL⁻¹ DNase I at 37 °C for 30 min to collect single cell suspension. Although we can not rule out that this may lead to unphysiological shifts in cellular distribution, all groups underwent the same procedure, which might counteract the adverse effects. Similar works were found in some papers (*ACS Nano*, 2017, 11: 9536-9549; *Nat Biomed Eng*, 2019, 3:729-740). Furthermore, according to the reviewer’s suggestion, we observed intuitively the colocalization of MPs or Man-MPs with immune cells in tumor tissues and got the same conclusion.

Besides CD206, CD209 (a C-type lectin) can also bind mannose residues. To determine whether Met@Man-MPs were mainly targeted to M2-like TAMs by binding with CD206, the colocalization of PKH26-labelled MPs or Man-MPs (red) with CD206- or CD209-positive cells (labelled with Cy3-conjugated CD206 or

CD209 antibody, respectively; green) in tumors, livers, spleens or lungs of H22 tumor-bearing mice at 24 h after intravenous injection of PKH26-labelled MPs or Man-MPs was observed by confocal microscopy. Our results showed that Man-MPs was mainly colocalized with CD206⁺ cells in tumor tissues, confirming the specific targeting of Man-MPs to CD206⁺ cells in tumors. We added the supplemented the data in the revised manuscript, Supplementary Figure 9.

(2). Tumor implantation site has profound effect on tumor status, including on vascular function and leakiness (for example, with orthotopic models typically showing more normal and less leaky blood vessels). Authors should provide rationale for their using the s.c. H22 model and perform at least a minimum set of experiments to show the validity of their observations in orthotopic model.

Response: We thank the reviewer's constructive suggestion. In the original manuscript, we evaluated the anticancer activity and improved tumor immune microenvironment induced by Met@Man-MPs in murine H22 subcutaneous tumor model and 4T1 orthotopic breast tumor model. Meanwhile, the synergistic anticancer activity of Met@Man-MPs and anti-PD-1 antibody was also evaluated in these two tumor-bearing mouse models. In the revised manuscript, we supplemented the azoxymethane (AOM)/dextran sodium sulfate (DSS)-induced colitis-associated cancer (CAC) mouse model to further evaluate the synergistic anticancer activity and improved tumor immune microenvironment induced by combination of Met@Man-MPs and anti-PD-1 antibody. We added the supplemented data in the revised manuscript, Figure 9 and Supplementary Figures 38-40.

(3) Marimastat is a first generation of MMP inhibitor based on the structure of the collagen molecule. It should be mentioned that hydroxamate activity-based MMP probes related to marimastat bind many off-target metalloproteinases that are not MMPs (Proc Natl Acad Sci USA. 2004;101:10000–10005).

Response: We thank the reviewer's constructive suggestion. In the Discussion section of the revised manuscript, we mentioned that batimastat (Bati), like its derivative marimastat having a collagen-mimicking hydroxamate structure, can bind off-target metalloproteinases that are not MMPs. Considering that Bati might not inhibit MMP specifically, we commented that besides MMPs, whether other metalloproteinases are involved in the biological function of Met@Man-MPs remains to be elucidated in the Discussion section of the revised manuscript.

(4) The figures are very busy and would benefit from moving nonessential graphs into the SI section.

Response: We thank the reviewer's constructive suggestion. We adjusted the formats of some figures and also moved some nonessential graphs to the SI section in the revised manuscript.

Other points.
Introduction

(5) L95-96 statement “macrophage derived MPs have natural tumor targeting capacity” is not supported by references used to support the statement. Ref 37 is a review paper that does not mention microparticles and Ref 38 deals with development of exosome-like macrophage-derived nanovesicles.

Response: We thank the reviewer for pointing out the mistake. We replaced them with the right reference in the revised manuscript.

(6) Discussion on translational potential of microparticles (including large scale manufacturing feasibility) and ongoing clinical trials needs to be included.

Response: We thank the reviewer’s constructive suggestion. We added some discussion on the possible improvement of the technology and translational potential of Met@Man-MPs in the Discussion section in the revised manuscript.

(7) The authors should avoid talking generally about “mannose receptor” (e.g. in L96), L137) and always mention the receptor type. Mannose and its analogues are not specific for CD206: they also bind other mannose receptors, such as CD209 expressed in the skin and intestinal and genital mucosa.

Response: We thank the reviewer’s constructive suggestion. Due to high expression of mannose receptor CD206 in M2-like macrophages, we emphasized that the M2-like macrophage targeting ability of Man-MPs was mediated by mannose receptor CD206 expressed in M2-like macrophages.

Figures and Results:

(8) Fig. 1.

- A – Include labeling of antibody: “Anti-PD1”

Response: We added the labeling of anti-PD-1 antibody in Figure 1A in the revised manuscript.

- D - Include lower mag views to allow appreciating the larger population of MPs.

Response: We thank the reviewer’s constructive suggestion. We put the TEM images with lower magnification to observe the larger population of MPs in the revised manuscript.

(9) Fig. 3.

- F-H - Start Y axis at 40% viability for better visualization of the effect.

Response: We thank the reviewer’s constructive suggestion. We modified Figure 3H-F by starting Y axis at 40% viability in the revised manuscript.

(10) Fig. 5.

- A – should be moved to SI. It is unclear why confocal is used in this cell free assay. Collagen layer appears non-uniform making one to question the assay and think of alternatives (such as double quenched collagen assays). Epifluorescence microscopy and using higher magnification to better see the MPs would give better understanding of fate of collagen layer upon incubation with the MPs. Labeling should be used to

point to areas of degradation. High dose MP samples should be treated also with optional inhibitors of MMPs and serine proteases (as it is likely that serine proteases such as plasminogen activation system is involved in the activation of collagen degrading proteases).

Response: We thank the reviewer's constructive suggestion. In the revised manuscript, we optimized the condition of collagen I polymerization. Collagen I was adjusted to pH 7.4 by adding NaOH, and then polymerized at 37 °C for 1 h and dried overnight to form uniform collagen I films. To evaluate whether MPs and Man-MPs can degrade tumor ECM by MMP activity, PKH26-labelled MPs or Man-MPs pretreated with or without batimastat (Bati, a broad-spectrum MMP inhibitor) or 4-(2-Aminoethyl)benzenesulfonyl fluoride hydrochloride (AEBSF, an inhibitor of serine proteases) were incubated with collagen I film. Collagen I was efficiently degraded by MPs or Man-MPs as evidenced by no colocalization between collagen I and MPs or Man-MPs. Pretreatment with Bati or AEBSF significantly decreased MPs- and Man-MP-induced collagen I degradation, suggesting that MMP was involved in the MPs- and Man-MPs-induced collagen degradation. Here we used confocal microscope to observe the overlay of PKH26-labelled MPs or Man-MPs and collagen I because using confocal microscope makes the background cleaner and the signal-to-noise ratio higher, generating relatively high quality images. Meanwhile, higher magnification was used to better observe the overlay of PKH26-labelled MPs or Man-MPs and collagen I in the revised manuscript. We added the supplemented data in the revised manuscript, Supplementary Figure 30.

- B – Early time point should be moved to SI, it adds clutter with minimal additional information. High-dose MP samples should be treated also with optional inhibitors of MMPs and serine proteases (as it is likely that serine proteases such as plasminogen activation system is involved in the activation of collagen degrading proteases).

Response: We thank the reviewer's constructive suggestion. In the revised manuscript, soluble collagen I was incubated with MPs or Man-MPs pretreated with or without batimastat (Bati, a broad-spectrum MMP inhibitor) or 4-(2-Aminoethyl)benzenesulfonyl fluoride hydrochloride (AEBSF, an inhibitor of serine proteases) for 72 h, and then the collagen content was measured by Sirius red total collagen detection kit. Our results also showed that MPs or Man-MPs efficiently degraded collagen in a MMP-dependent manner. According to the reviewer's suggestion, we moved the original Figure 5b to Supporting Information, Supplementary Fig. 31, and added the supplemented data in the revised manuscript, Figure 5a.

C,D – Generally, different tumor areas show greatly varying deposition of ECM and for conclusive data it is essential to include a low magnification view of the entire tumor. It is unclear if n=10 refers to independent tumor mice or fields analysed, please specify.

Response: We thank the reviewer's constructive suggestion. Second-harmonic generation (SHG) microscopy was used to analyze the fibrillar collagen structure in

tumors. In this work, we used an LSM 780 NLO upright microscope (Carl Zeiss AG, Germany) equipped with a W Plan-Apochromat 20×objective (the minimum magnification in this imaging system) to acquire SHG images. To acquire the images of larger areas, a wide range of continuous shooting mode was applied using Zen lite 2012 software from the Zeiss microscope and the images were then stitched together (**Figure 2**). The results intuitively showed that MPs, Man-MPs and Met@MPs significantly decreased the deposition of collagen fiber network, confirming the collagen-degrading capability of MPs and Man-MPs. Due to the stitching marks existed in the images, we did not use these images in the revised manuscript but kept the original images. We quantified 10 fields from 3 mice randomly, which can reflect the overall trends of fibrillar collagens in different groups.

Figure 2. SHG imaging of tumor tissues of H22 tumor-bearing mice after intravenous injection of PBS, MPs, Man-MPs, free Met, Met@MPs or Met@Man-MPs at the Met dosage of 10 mg kg^{-1} , or high dosage of Met at 100 mg kg^{-1} every two days for different times. Scale bars: $200 \mu\text{m}$.

- E,F – Same comment and for CD on magnification – to be convincing, larger areas need to be shown. Authors should comment on collagen1 staining pattern and lack of fibrillary structures.

Response: We thank the reviewer’s constructive suggestion. Collagen I staining in tumor tissues was performed after the tumors were sectioned by cryotomy, stained with Cy3-conjugated collagen I antibody and then observed by fluorescence microscope. In the original manuscript, the brightness and contrast were not adjusted properly when taking pictures using fluorescence microscope. In the revised manuscript, we adjusted the parameters of the fluorescence microscope and supplemented the images with a low magnification view. We added the supplemented data in the revised manuscript, Figures 5d and 5e.

(11) Fig. 6

- B –indicate the tumor margin by a dotted line and add additional tissue context by performing CD31 staining (to allow relating CD8⁺ cell and vascular densities. A low mag view of entire tumor with CD8⁺ cells highlighted would be more convincing.

Response: We thank the reviewer's constructive suggestion. In the revised manuscript, tumor tissues were labelled with Cy3-conjugated CD8 antibody (labelling CD8⁺ T cells), FITC-conjugated CD31 antibody (labelling tumor vascular) and DAPI (labelling nuclei) and then observed by fluorescence microscope. The immunofluorescence images with a low magnification view were acquired. The tumor margin was marked by a white dotted line in these images. We added the supplemented data in the revised manuscript, Figure 6b.

- C,D – authors should comment on surprisingly smaller effect of Met@Man-MPs on inducing CD8⁺ cell number in tumors in histology based analysis of tumor interior (C, left) compared to total tumor (D). It would be expected that the difference is larger at the center, as it is not diluted by margin with no difference.

Response: We thank the reviewer's insightful question. In this work, the ordinate of Figure 6C was the CD8⁺ T number per field based on the total fluorescence of CD8⁺ T cells per field by confocal microscopy, while the ordinate of Figure 6D was the CD8⁺ T cell number measured by flow cytometry divided by tumor weight (the ordinate unit was CD8⁺ T cell number/g tumor). Due to the good anticancer activity of Met@Man-MPs, the tumor weight of Met@Man-MPs-treated group was significantly decreased (about 25% of the PBS-treated group), which magnified the difference between the PBS-treated group and Met@Man-MPs-treated group compared with the data in Figure 6C.

(12) Fig. 7

Here and elsewhere would be reader-friendly to use color-coded text to explain labeling.

Response: We thank the reviewer's constructive suggestion. We used color-coded text to explain the labeling of figures in the revised manuscript.

- C - Accumulation of anti-PD1 antibody should be studied in the context of vascular permeability. At minimum, the tumors should be stained for endogenous mouse IgG as a surrogate marker as vascular permeability.

Response: We thank the reviewer's constructive suggestion. In this work, we tried to determine the enhanced tumor penetration of anti-PD-1 antibody induced by Met@Man-MPs. The colocalization of Cy5-labelled anti-PD-1 antibody (red) with endothelial cells labelled with FITC-conjugated CD31 antibody (green) was observed in tumor sections of H22 tumor-bearing mice at 24 h after treatment with Cy5-labelled anti-PD-1 antibody and Met@Man-MPs pretreated with or without batimastat (Bati, a broad-spectrum MMP inhibitor). Our results showed that anti-PD-1 antibody was strongly colocalized with tumor vessels due to the dense tumor extracellular matrix, although the tumor vessels were permeable as evidenced

by the fact that endogenous IgG was distributed throughout the entire tumor tissues (Supplementary Figure 34 in the revised manuscript). Met@Man-MPs treatment resulted in the wide distribution of anti-PD-1 antibody in tumor tissues, suggesting that anti-PD-1 antibody was extravasated from tumor vessels and then penetrated into tumor tissues. However, the improved tumor penetration of anti-PD-1 antibody was compromised in Bat-pretreated Met@Man-MPs group, further confirming that Met@Man-MPs with MMP activity-induced collagen degradation contributed to the tumor penetration.

- D – number of mice used to generate the data points needs to be specified (information on number of fields is provided).

Response: The data points generated in Figure 7D was from 15 fields for three mice. We added the detailed information in the revised manuscript.

- E – Include labeling (arrows) to point to extravascular PD1 signal. It is not mentioned that lectin was used as a vascular tracer, add.

Response: We thank the reviewer's constructive suggestion. We added the arrows to point out the extravascular PD-1 signal in Figure 7E in the revised manuscript. Meanwhile, we added the information about vascular labeling using fluorescein-labelled tomato lectin in the figure legends of Figure 7E.

- F - number of mice used to generate the data points needs to be specified.

Response: The data points generated in Figure 7F was from 10 fields for two mice. We added the detailed information in the revised manuscript.

Materials and methods

- L619-620: please specify how were the metastatic nodules counted? On tissue sections based on H&E stain, macroscopically?

Response: In this work, the lungs of 4T1 tumor-bearing mice were fixed in Bouin's solution overnight at room temperature, and then the metastatic tumor nodules in the lungs were counted macroscopically. We added the details in the Methods section in the revised manuscript.

- L723: Is it correct that tumor slices for organotypic culture were 5 mm (0.5 cm) thick? This is very thick and poses a serious diffusion problem for the gas and nutrient exchange. Authors should correct and/or comment on that.

Response: We thank the reviewer's insightful question. In the original manuscript, we did not state clearly the volume of tumor slices, just like "tumors were cut into thin slices of about 5 × 5 mm". In fact, the thickness of tumor slices was 2 mm. We supplemented the details of tumor slices as about 5 × 5 × 2 mm in the revised manuscript.

REVIEWERS' COMMENTS

Reviewer #1 (Remarks to the Author):

After reading the rebuttal letter and the new data present in the revised manuscript, I am overall satisfied by the answer and new added data.

I therefore suggest the acceptance of the manuscript

Reviewer #2 (Remarks to the Author):

The revised manuscript by Drs Wei et al. entitled "Boosting Anti-PD-1 Therapy by Metformin-Loaded Targeting Microparticles Derived from Macrophages" has been greatly improved for clarity and academic presentation. Compared to original report the text and figures are easier to follow and intuitive. Important details on reproducibility have been answered and mistakes corrected. The outcome is a comprehensive and well written report on an important field that I enthusiastically recommend for publication.

Response to Reviewer 1

After reading the rebuttal letter and the new data present in the revised manuscript, I am overall satisfied by the answer and new added data. I therefore suggest the acceptance of the manuscript.

Response: Thank you very much for the reviewer's comments. We are very happy that the revision of this manuscript satisfied you.

Response to Reviewer 2

The revised manuscript by Drs Wei et al. entitled “Boosting Anti-PD-1 Therapy by Metformin-Loaded Targeting Microparticles Derived from Macrophages” has been greatly improved for clarity and academic presentation. Compared to original report the text and figures are easier to follow and intuitive. Important details on reproducibility have been answered and mistakes corrected. The outcome is a comprehensive and well written report on an important field that I enthusiastically recommend for publication.

Response: Thank you very much for the reviewer’s comments. We are very happy that the revision of this manuscript satisfied you.